# A Novel Method for Improving Air Pollution Prediction Based on Machine Learning Approaches: A Case Study Applied to the Capital City of Tehran

**Mahmoud Reza Delavar** [1,*] **, Amin Gholami** [2] **, Gholam Reza Shiran** [3] **, Yousef Rashidi** [4] **, Gholam Reza Nakhaeizadeh** [5] **, Kurt Fedra** [6] **and Smaeil Hatefi Afshar** [2]

[1] Center of Excellence in Geomatic Engineering. in Disaster Management, School of Surveying and Geospatial Engineering, College of Engineering, University of Tehran, P.O. Box 1439951154 Tehran, Iran

[2] Department of GIS, School of Surveying and Geospatial Engineering, College of Engineering, University of Tehran, P.O. Box 1439951154 Tehran, Iran; amingholami328@gmail.com (A.G.); hafshar@ut.ac.ir (S.H.A.)

[3] Dept. of Transportation Eng., Faculty of Civil & Transportation Engineering, University of Isfahan, P.O. Box 8174673441 Isfahan, Iran; gholam_shiran@yahoo.com

[4] Environmental Scienecs Research Institute, Shahid Beheshti University, P.O. Box 1983969411 Tehran, Iran; y_rashidi@sbu.ac.ir

[5] APL-Professor of Economics and Econometrics, Karlsruhe Institute of Technology, Institute of Economics Econometrics and Statistics, 76049 Karlsruhe, Germany; gholamreza.nakhaeizadeh@kit.edu

[6] Environmental Software & Services GmbH., A-2352 Vienna, Austria; kurt@ess.co.at

* Correspondence: mdelavar@ut.ac.ir; Tel.: +98-21-6111-4257

**Abstract:** Environmental pollution has mainly been attributed to urbanization and industrial developments across the globe. Air pollution has been marked as one of the major problems of metropolitan areas around the world, especially in Tehran, the capital of Iran, where its administrators and residents have long been struggling with air pollution damage such as the health issues of its citizens. As far as the study area of this research is concerned, a considerable proportion of Tehran air pollution is attributed to $PM_{10}$ and $PM_{2.5}$ pollutants. Therefore, the present study was conducted to determine the prediction models to determine air pollutions based on $PM_{10}$ and $PM_{2.5}$ pollution concentrations in Tehran. To predict the air-pollution, the data related to day of week, month of year, topography, meteorology, and pollutant rate of two nearest neighbors as the input parameters and machine learning methods were used. These methods include a regression support vector machine, geographically weighted regression, artificial neural network and auto-regressive nonlinear neural network with an external input as the machine learning method for the air pollution prediction. A prediction model was then proposed to improve the afore-mentioned methods, by which the error percentage has been reduced and improved by 57%, 47%, 47% and 94%, respectively. The most reliable algorithm for the prediction of air pollution was autoregressive nonlinear neural network with external input using the proposed prediction model, where its one-day prediction error reached 1.79 µg/m³. Finally, using genetic algorithm, data for day of week, month of year, topography, wind direction, maximum temperature and pollutant rate of the two nearest neighbors were identified as the most effective parameters in the prediction of air pollution.

**Keywords:** air pollution; prediction; machine learning; regression SVM; geographically weighted regression; artificial neural network; auto-regressive nonlinear neural; interpolation; genetic algorithm

## 1. Introduction

Air pollution is one of the most important environmental issues in both developed and developing countries. Air pollution means the existence of one or more pollutants contaminating outdoor or indoor air in various amounts and periods which may harm human, vegetation or animal life or unexpectedly interacts with normal life or properties [1,2]. The distribution of air-pollution involves a complex process depending on a number of factors. In fact, air pollution prediction, which has a non-linear dynamism, is a very difficult task and requires a close understanding of the dispersion of air pollutants in the atmosphere, which involves an immense cost [3]. In some cases, air pollution in mega cities even exceeds the standard limit which increases the concerns. For this reason, air pollution has become a problem in many cities in the world and its investigation is considered as a vital issue in urban management. The general sensitivity towards this problem has urged the officials to pass laws in order to prevent the air-pollution [4]. One of urban managers' objectives is to provide the citizens with the right information to make them aware of air quality rates [5]. The pollution information includes the density of daily $PM_{2.5}$ and $PM_{10}$ pollutants which can be announced to the concerned people by city managers as a response to the air pollution [6]. This information may assist people to avoid the polluted areas and employ public transport facilities to reduce the level of the pollution. In addition, the concerned city managers can implement the information to control the urban traffic and the responsible pollutant industries and to increase public transport facilities in order to mitigate the level of the pollution. To achieve this goal, appropriate tools need to be used to predict air pollution [6].

According to the latest available statistics from 21 stations belonging to Tehran Air Quality Control Company (AQCC) and 16 air-pollution measurement stations belonging to the Iranian Environmental Protection Agency, $PM_{10}$ and $PM_{2.5}$ constitute the highest proportion concentration of air-pollution in Tehran. Among the pollutants such as CO, $O_3$, $NO_2$, $SO_2$, $PM_{10}$ and $PM_{2.5}$, $PM_{2.5}$ has the highest share. Based on the studies undertaken in 2017 by AQCC and the technical report produced on the Tehran Air Pollution Prediction System, nearly 5% of $PM_{2.5}$ pollutants are coming from neighboring populated areas laid in the west (city of Karaj, south west of Tehran (city of Shahryar), and south east of Tehran (city of Rey)). Such a percentage has been found higher in the summer time due to higher levels of wind speed in transporting the dust driven from out west and trapped in the Greater Tehran basin [7]. The percentage presented here on $PM_{2.5}$ pollutant can be regarded, based on the AQCC expert opinion, as the highest percentage with respect to other pollutants that have been detected to be under 5%. Furthermore, the $PM_{2.5}$ detected in the winter time above is not of natural or wind-blown dust from outside deserts [7].

$PM_{2.5}$ contaminants contain particles that are created by combustion or caused by the formation and compression of secondary particles. $PM_{10}$ particles contain particles that are 10 micrometers in diameter and smaller and can pass through the first defensive barrier (nose and throat), damage the lungs and deposition there [8]. Studies have shown that exposure to suspended particles is associated with health effects such as cardiovascular and respiratory diseases [9]. The World Health Organization estimates that if the average annual concentration of $PM_{10}$ is reduced from 70 μg/m$^3$ to 20 μg/m$^3$, then the associated deaths will be reduced by 15% [10]. In fact, there is a relationship between the exposure to intense concentrations of suspended particles and the increase in daily and annual mortality, as well if the concentration of these pollutants is reduced while other factors are fixed then the associated deaths are reduced [10]. These particles are very tiny and their damage to human health is high. In this study, $PM_{2.5}$ and $PM_{10}$ are used as pollutants to predict air pollution. Hence, air-pollution prediction is becoming one of the managerial solutions to prevent and/or mitigate its destructive implications. Therefore, it seems necessary to predict $PM_{10}$ and $PM_{2.5}$ pollutants using the appropriate methods.

In the past few decades, two general approaches of deterministic and stochastic methods have been used to predict air-pollution [11]. Diffusion models are among the deterministic methods developed in various regions for modeling and monitoring the air pollution [12,13]. However,

the output of these models relies on the input data, and in order to use them, it is necessary to access the data on how the pollutants disseminate and diffuse in the atmosphere [14].

Therefore, using these models where sufficient and precise data is not accessible is problematic. Considering that the data collection needed for diffusion models is very hard and impossible at large scales, the researchers have turned to superior methods such as statistical models [15]. Compared to the deterministic methods, statistical methods have more application in prediction of air-pollution. It is worth mentioning that factors such as air pressure, temperature, humidity, rainfall and wind affect the pollutants dissemination [16].

A study has been conducted by [17] with the aim of predicting the density of two pollutants (CO and NOx) in industrial locations using the autoregressive model based on artificial neural network using some meteorological parameters. As a result of performance of the proposed model, Root Mean Square Error (RMSE) for CO and NOx pollutants was 0.8445 and 0.7618, and the mean absolute error (MAE) for the pollutants was 0.1451 and 0.1598, respectively. The results show the higher importance of meteorology variables in the prediction of pollutant concentration and the efficiency of the neural network in the air pollution prediction.

The authors of [18] introduced a model to improve the artificial neural network, which is a combination of air mass route analysis and wavelet transform. The rate of RMSE for the combinational model can be decreased by 40% on average. The study verified that especially on the days with a higher concentration of $PM_{2.5}$ often predicted for the warned threshold of the combinational models using wavelet analysis and detection rate (DR), the RMSE can reach to the average limit of 90%. This approach indicates the potential of the proposed model in air-quality prediction system in other countries.

With the aim of time series analysis in Abura region in Colombia, in different temporal scales (daily, weekly, and yearly), the geostatistical methods were suggested by [19] so as to use the obtained information for calculation of unknown air quality values and prediction of air pollution. According to the results from the proposed method for prediction of $PM_{2.5}$ concentration on a daily basis, the amount of correlation of coefficient ($R^2$) that was obtained is equal to 0.55.

The authors of [20] have used two methods of land use regression and Universal Kriging to predict the concentration of NOx in the city of Los Angeles. In addition to using the meteorological and pollutant concentration parameters, spatial parameters such as roads, population, land use and distance from the coastal regions were used. The results suggest that in prediction of NOx concentration, the universal Kriging model has more precision than land use regression. The authors of [21] have carried out a one-year analysis of ozone concentration in the Malaga region of Spain. The multivariable regression for prediction of ozone concentration employing the meteorology parameters was employed. Diffusion models and statistical methods such as Kriging in modeling the air pollution face some limitations. The output of diffusion models is highly associated with input data and it is necessary that the data with high precision are available about the way the pollutants diffuse and disseminate in the atmosphere [5]. Though the common statistical models of Kriging have also been used for spatial modeling of air-pollution, its ratio is constant relative to the temporal variations [15,22]. That is why in recent years the machine learning methods have been of interest to researchers [5]. The authors of [23] have employed neural networks for air-pollution prediction. The correlated parameters with the air pollutant include traffic, hours and days of week, pollutant concentration in the past 3 years, the wind speed and direction, temperature, solar radiations, rainfall, relative humidity rate and the distance from the road. The authors of [24] have used the nonlinear autoregressive exogenous (NARX) Neural Network model for prediction of time series of ozone concentration peak. The results showed that this kind of neural network had a good performance in time series prediction of ozone concentration in Milan.

Support vector machine (SVM) and partial least square (PLS) method have been implemented for prediction of CO concentration in Rey station in Tehran [25]. The data related to $O_3$, $SO_2$, NOx, $CH_4$, total hydrocarbons (THC) and meteorological data such as air pressure, temperature, wind speed and direction, and air humidity were used in a period of January 2007 to January 2011. The results verified

the good accuracy of both the methods, while the combinational method of PLS-SVM is faster and better than support vector model.

The above-mentioned researches have used different machine learning algorithms which are the statistical methods for prediction of air pollution. In this research, the supervised algorithms for machine learning regression such as artificial neural network (ANN), the nonlinear autoregressive exogenous Neural Network, geographically weighted regression (GWR) and support vector regression (SVR) were used to predict $PM_{2.5}$ and $PM_{10}$ pollutants.

In general, two types of methods including satellite imagery and ground sensors are used to collect air pollution data. Given the cost, availability and accuracy of ground sensor data for 10 years, this type of data has been used in this research. One of the major problems of ground-based sensors is the calibration of the device. The air pollution data have been validated by Tehran Air Quality Control Company (AQCC). In spite of the above assumption, in this study, a data refinement mechanism was used to better estimate the concentration of pollutants in areas where there is a data gap or an error in the data.

To the best of our knowledge, no report is published so far regarding input data refinement for network learning, and this research seeks to improve the accuracy of these methods and select the best one for air pollution prediction. In addition, there have been few studies concerned with the identification of effective parameters in prediction of air-pollution based on statistical models, which is one of objectives of this research.

The aims of this research are as follows:

- Selecting the best statistical model and its improvement for air-pollution prediction;
- Selecting the best refinement method for air-pollution and meteorological data in order to predict the missed data and filter the noise of the data; and
- Determination of the most influencing parameters in air pollution prediction.

To generate a dataset to be entered in the machine algorithm process, it is necessary first to interpolate the meteorological parameters for transfer of parameters from meteorology stations to air-pollution stations. Considering that there are only five meteorology stations in Tehran, the inverse distance weighting (IDW) method was assumed in this research to be suitable for interpolation of these parameters with a less number of control points.

The rest of the paper follows the materials and methods in Section 2, the results including the data preparation and refinement and air pollution prediction are presented in Section 3, discussion is presented in Section 4 and finally conclusions including major results, contributions and future directions are covered in Section 5.

## 2. Materials and Methods

Air pollution is a phenomenon, which is affected by different factors. In order for precise prediction, right identification of these parameters influencing the air pollution is necessary. For example, one of these parameters is the month of the year where the relationship between $PM_{2.5}$ contaminant concentration and the months of the year is plotted in Figure 1.

According to Figure 1, with the reduction of traffic and improvement of atmospheric conditions, the lowest monthly concentration of $PM_{2.5}$ contamination occurred in March which coincides with the New Year holidays in Iran. The monthly average of the pollutant concentration with the onset of the warm season and the occurrence of the dust phenomenon, dust has been risen in the city. Of course, the highest amounts of $PM_{2.5}$ monthly concentrations are observed in December and October, respectively, due to the air flow, the increase in atmospheric stability and the inversion of temperature, which has led to the accumulation of pollutants in the city [7].

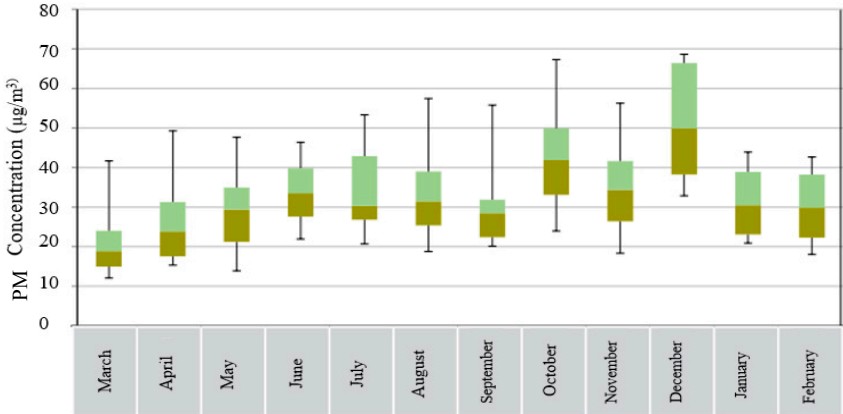

**Figure 1.** Monthly changes in PM$_{2.5}$ concentration in Tehran in 2016.

Another important parameter influencing the air pollution concentration is wind, which includes two components of wind speed and direction. Figure 2 illustrates the map of the Tehran wind speed and direction and its surrounding cities.

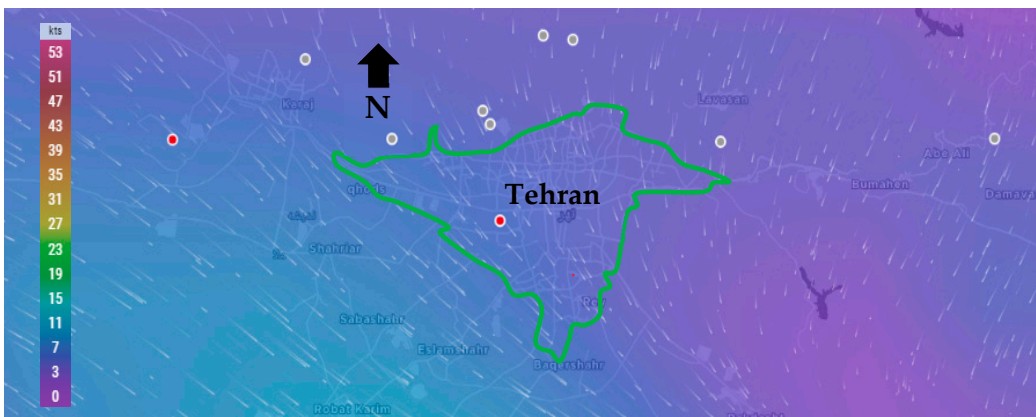

**Figure 2.** A map of the wind speed and direction of the city of Tehran and its surroundings (https://www.windfinder.com/#10/35.6841/51.3474/2019-02-12T18:00Z).

According to Figure 2, Tehran wind direction is from the northwest to the south-east. Therefore, the wind transfers air pollution from different parts of the city and even from cities located in the northwest of Tehran (such as Karaj) to the south and Southeast of the mega city.

All of the influential parameters can be divided into two groups of spatial and temporal data. Temporal data refers to those data, which rapidly change in a short duration of time. Effective parameters are parameters that changes in location and time which cause changes in the concentration of air pollution. Figure 3 illustrates the division of the employed data in the two groups of spatial and temporal data.

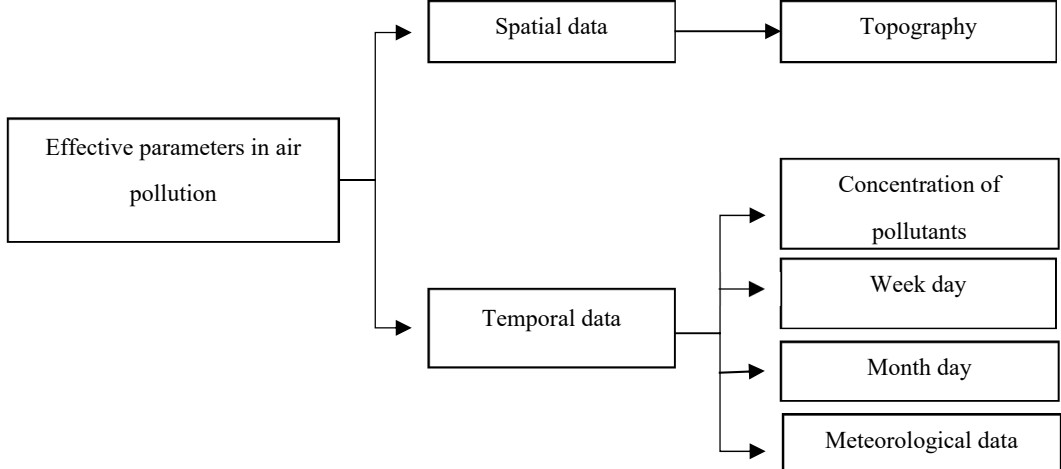

**Figure 3.** The data employed in this research.

The parameters influencing the air pollution include topographic components (x, y, z) due to vast area of Tehran and significant height changes; meteorological components due to their direct or indirect effects on pollutant emission; week-time components, as in various days of week, the traffic in Tehran is different and accordingly the amount of pollutant concentration will be different. For example, on the weekends, Tehran experiences less traffic density; and the month of year component also due to season changes, dust, and temperature inversion leads to changes in the extent of the pollutants. On the other hand, in this research, the air pollution values of the two nearest air pollution station neighbors are used as an influential parameter in the prediction of the air pollution.

The relevant data to concentration of $PM_{10}$ and $PM_{2.5}$ pollutants were obtained from AQCC during 2006 to 2016. The concentration of these pollutants is collected on a daily basis and delivered by the company (http://airnow.tehran.ir/home/dataarchive.aspx). Tehran municipality has 41 stations for measuring air pollution managed and controlled by Iranian Environmental Protection Agency and AQCC. As this research needs daily data which are unavailable for all the stations, only the data related to 24 stations have been used.

Meteorological data was provided by Iran Meteorological Organization in a 10-year period (2006 to 2016) from the Shemiran, Dowshan-Tappeh, Chitgar, Mehrabad Airport and Geophysics stations (http://irimo.ir/eng/index.php) which are presented in Section 3. These data include the maximum and minimum temperatures, wind speed and direction and humidity which have been collected on a daily basis. The meteorological and air pollution data have been quality controlled by the concerned organizations.

To predict the air-pollution, this research used machine learning methods such as SVR, GWR, artificial neural network, and NARX with external input. To this end, the meteorological data were employed for learning, collected from Iran Meteorological Organization during the ten year period. Given that these data were not error-free and there were lost data in these datasets for daily, weekly, and monthly durations, they were initially refined and prepared for machine learning. Afterward, all the methods were compared and the best model for air-pollution prediction was selected which is illustrated in Figure 4.

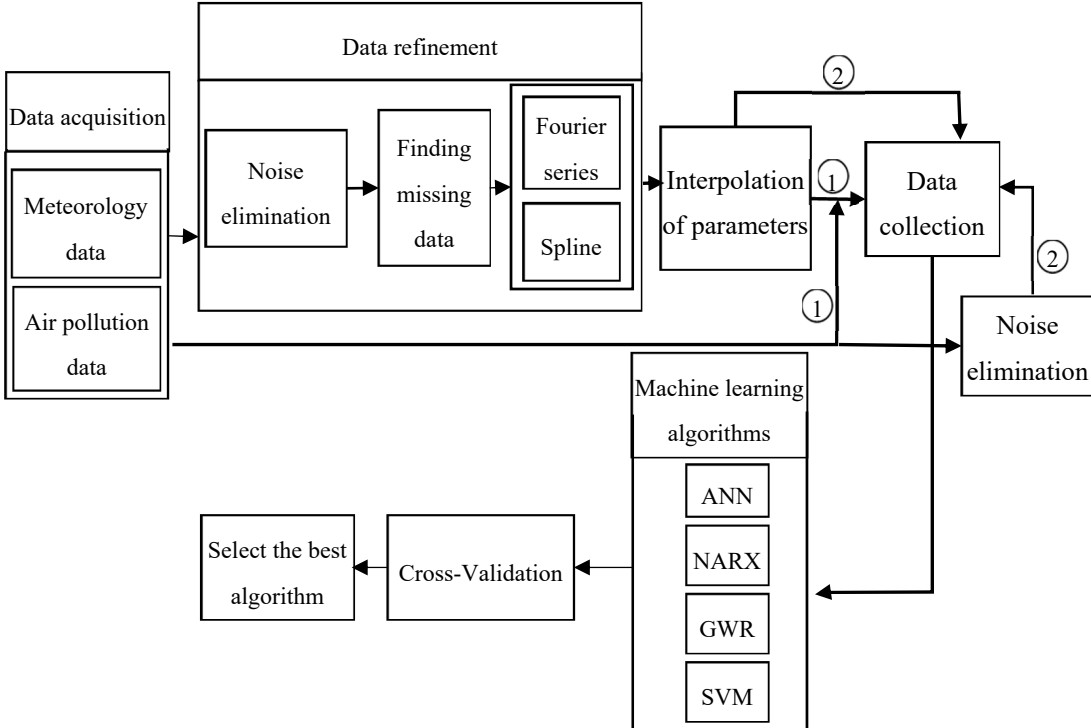

**Figure 4.** The proposed best air pollution prediction model.

As illustrated in Figure 4, the data related to meteorology and air-pollution has been obtained from meteorological organization and AQCC. These data are not error-free, requiring some algorithms to remove outliers and find the lost data. For this purpose, in this research, refinement and treatment of the data were conducted in several steps. Considering that the structure of time series for air pollutants could not be modeled in a precise form, only noise deletion algorithm has been run. This research uses two datasets for prediction of air-pollution. In the first dataset, the refined meteorological data and unrefined pollutant data, and in the second database, refined meteorological data and pollutant data have been used.

In order to remove the noise in the air-pollution and meteorological data, Savitzky-Golay filter was used. Savitzky-Golay smoothing filter (also referred to as smoothing polynomial filter or Least-Squares smoothing filter) is typically used to smooth a signal with high noise which has a long frequency (without noise). This filter is better than ordinary Finite Impulse Response (FIR) filter [26]. The Savitzky-Golay filter is optimum because it minimizes the least square error in multinomial connections to each frame of noise data [26]. The information obtained from the meteorological organization for the 10-year data was not complete. To compensate for this data gap, this research has used Fourier series and spline multinomial approaches. Having refined the meteorological data, values of these parameters need to be obtained in air-pollution stations to use this information in predicting the air-pollution. After these steps, the two datasets provided for running the air prediction algorithms were used. This research employs SVR, NARX, ANN, and GWR for air-pollution prediction, as summarized in the following section. Finally, the results were evaluated using cross-validation and the best method for modeling the air-pollution prediction was developed.

After each of the methods is implemented and modeled, the results should be validated and compared. Therefore, in this research, two parameters of the coefficient of determination and root mean square error using cross-validation method were used to evaluate the results. The determination coefficient shows the correlation between the observed values and the calculated values which is always between 0 and 1, the value of one represents a complete correlation between the observed values and the calculated values, and the zero value represents the independence of the observed

values and the values c alculated. The coefficient of determination and the root mean square error have been calculated using Equations (1) and (2) [27,28].

$$R^2 = \left[ \frac{1}{N} \frac{\sum_{i=1}^{N} \left[ \left( P_i - \overline{P} \right) \left( O_i - \overline{O} \right) \right]}{\sigma_p \sigma_o} \right]^2 \tag{1}$$

$$RMSE = \left( \frac{1}{N} \sum_{i=1}^{N} [P_i - O_i]^2 \right)^{\frac{1}{2}} \tag{2}$$

where $N$ is the number of observations, $O_i$ is the observed parameter, $P_i$ is the calculated parameter, $\overline{O}$ is the mean of the observation parameter, $\overline{P}$ is the average calculation parameter, $\sigma_o$ is the standard deviation of the observations and $\sigma_p$ is the standard deviation of the calculation.

## 3. Results

Tehran mega city, as the capital of Iran is the most important metropolis and the political and commercial center of the country with 10% of the country's total population. In 2016, Tehran's population was 8.6 million and Iran's population was 81.1 million. Tehran is surrounded by the mountain ranges of Alborz and the Kavir desert from north and south, respectively. Tehran's climate is affected by its geographical position. Air-pollution in Tehran has exceeded the standard limit. Therefore, prediction of air pollution is imperative for Tehran to allow for taking necessary measures for air pollution control and identification of the locations where air pollution level is in its dangerous level. In addition, it is possible to prevent the expansion of diseases originated from air pollution by informing the public, especially the highly sensitive air pollution people. For this purpose, in order to predict the air pollution, Tehran mega city was selected as the study region. Figure 5 shows the location of the Tehran air and weather stations used in this study.

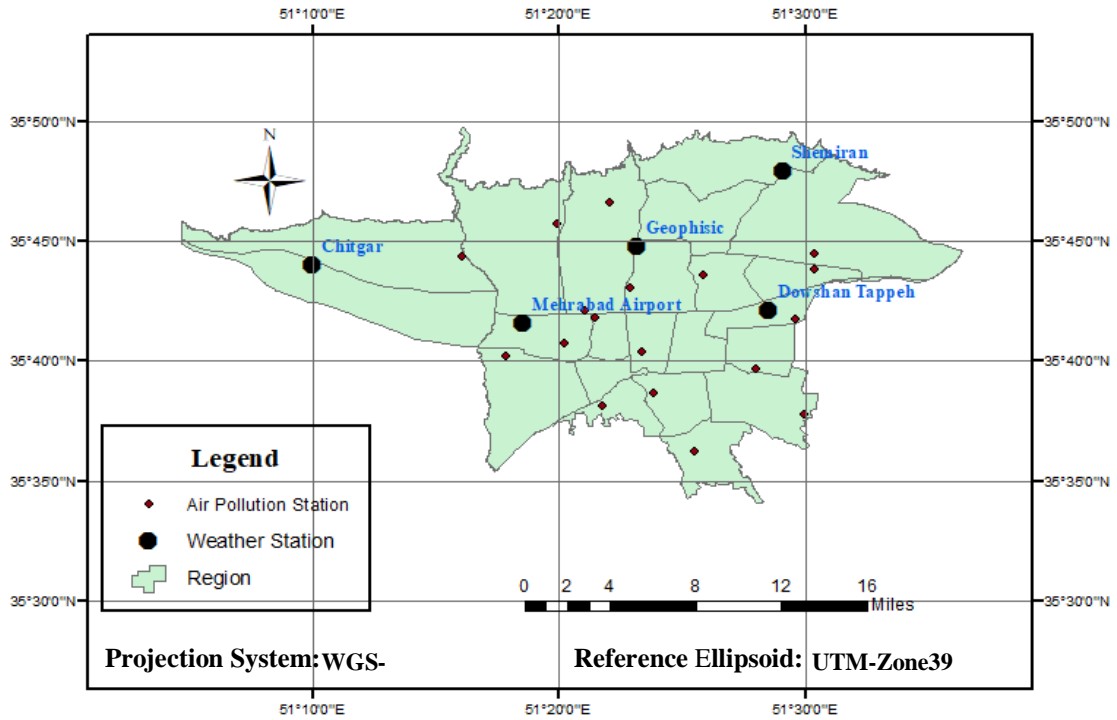

**Figure 5.** Tehran air pollution and weather stations.

## 3.1. Data Preparation and Refinement

In this phase, data preparation and refinement related to meteorology and air pollution are explained. In Figures 6–11, the daily time series of parameters acquired from meteorological stations including the wind speed and direction, maximum and minimum temperature, humidity and air pressure for Chitger station illustrated in Figure 5 are presented as examples of the total data collected.

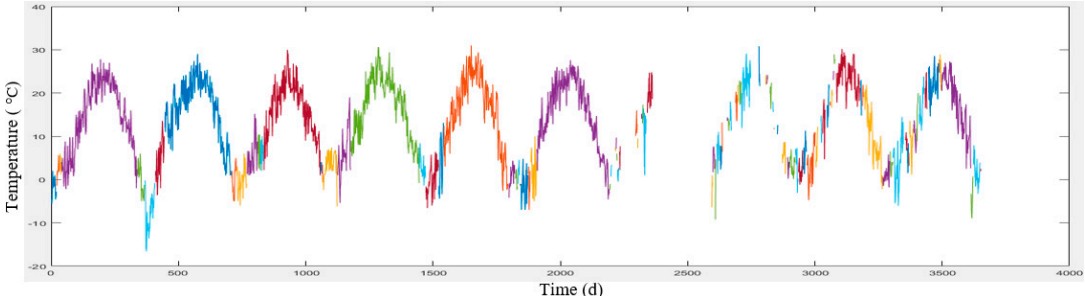

**Figure 6.** Chitgar station minimum temperature time series.

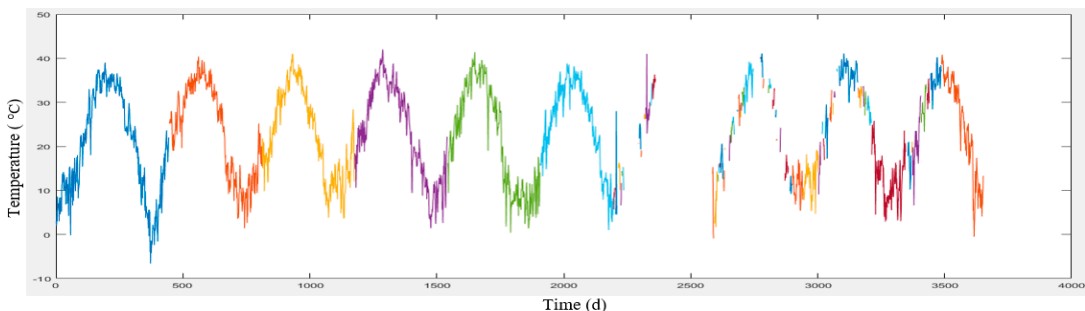

**Figure 7.** Chitgar station maximum temperature time series.

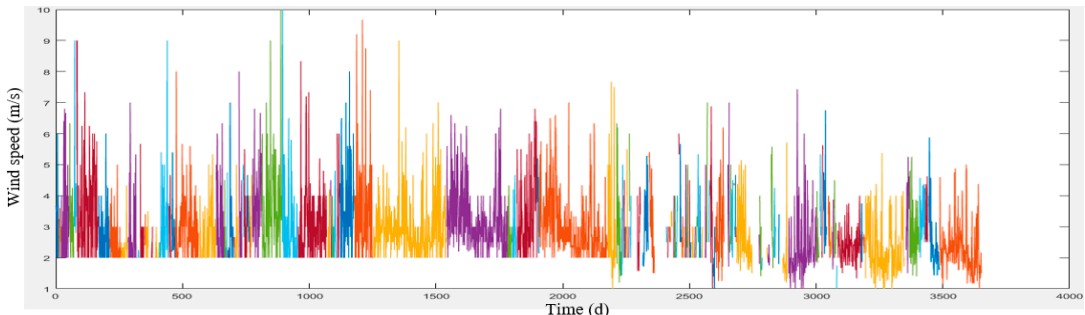

**Figure 8.** Chitgar station wind speed time series.

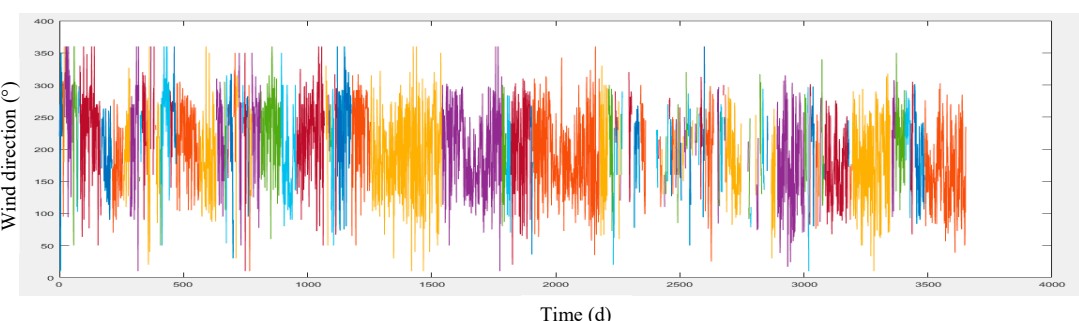

**Figure 9.** Chitgar station wind direction time series.

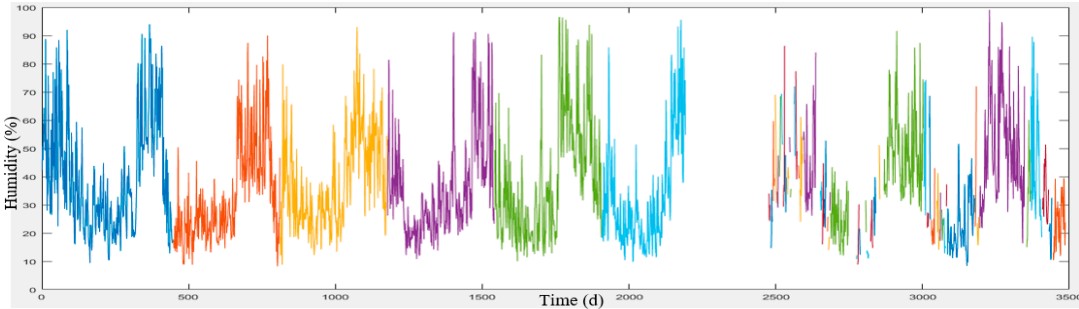

**Figure 10.** Chitgar station humidity time series.

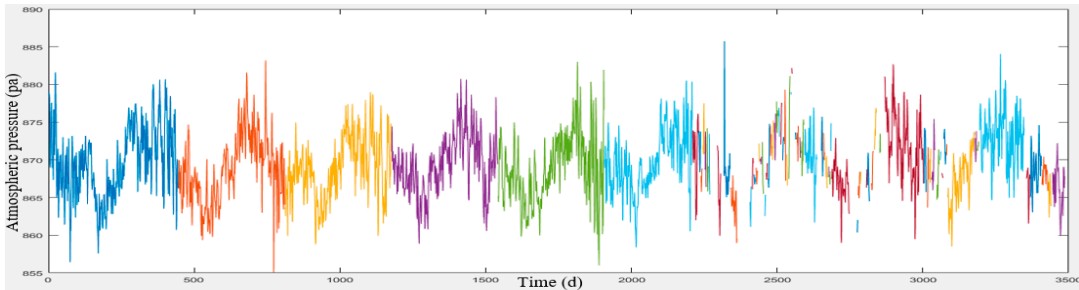

**Figure 11.** Chitgar station air pressure time series.

In Figures 6–11, the color change indicates that there are some missing data. As shown in these figures, Fourier series and spline cannot be individually modeled over the points as the data are erroneous and do not have a suitable structure for curve fitting. For this reason, firstly, the noise should be removed, where in this research a Savitzky-Golay filter is used. Considering the shape of sinus waves and wavelets, it can be mentioned that where the waves are irregular or have abrupt changes, they can be better analyzed using the Savitzky-Golay filter. Therefore, as the meteorological parameters are subject to abrupt changes and their signals are irregular during a period, the Savitzky-Golay filter has a better functionality.

Tehran meteorological and air pollution data have some missing values for $PM_{10}$ and $PM_{2.5}$ pollutants during the ten years period (2006 to 2016). Fourier series and splines have been used to remove the missing data for the meteorological data which is used as a training data in the machine learning process. Otherwise, less data would be available for the training of the machine learning process. In this research, the selection of a period when both air pollution data and all the intended weather parameters were simultaneously available was necessary. The input parameters of the Savitzky-Golay filter include the polynomial order and frame length whose best fit for order = 3 and frame length = 23 was obtained.

Figures 12–17 present the real-time series and the time series with noise deletion of the employed parameters. As indicated, Savitzky–Golay filter has a best fit in some points with abrupt changes.

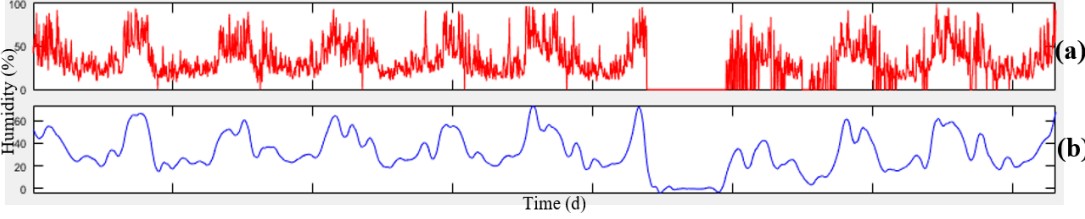

**Figure 12.** Time series (**a**) unrefined (**b**) refined humidity in Chitgar Station.

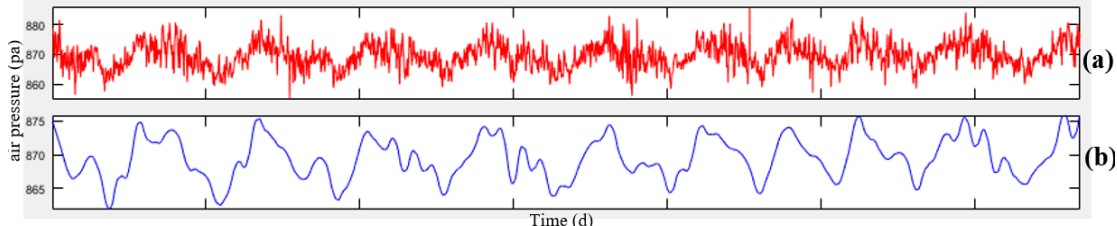

**Figure 13.** Time series (**a**) unrefined (**b**) refined air pressure in Chitgar Station.

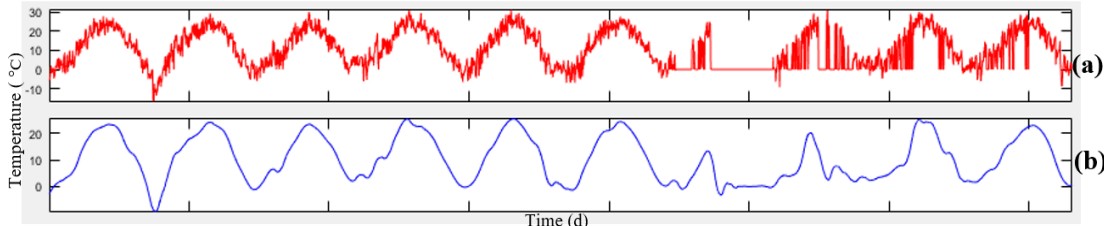

**Figure 14.** Time series (**a**) unrefined (**b**) refined minimum temperature in Chitgar Station.

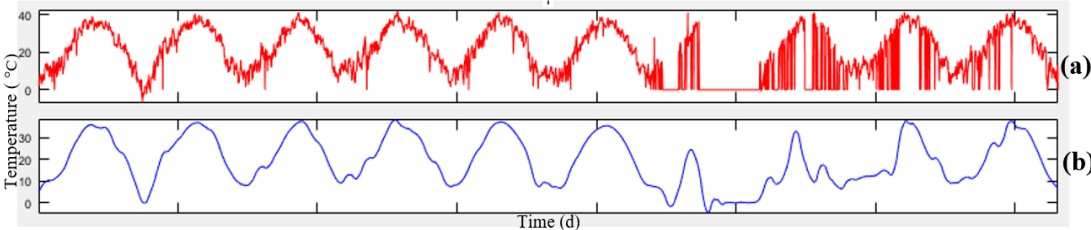

**Figure 15.** Time series (**a**) unrefined (**b**) refined maximum temperature in Chitgar Station.

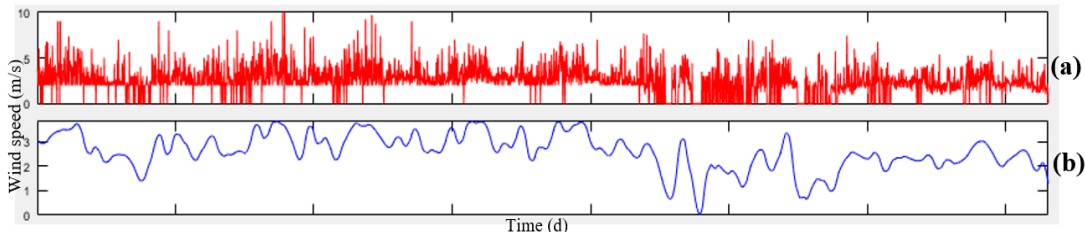

**Figure 16.** Time series (**a**) unrefined (**b**) refined wind speed in Chitgar Station.

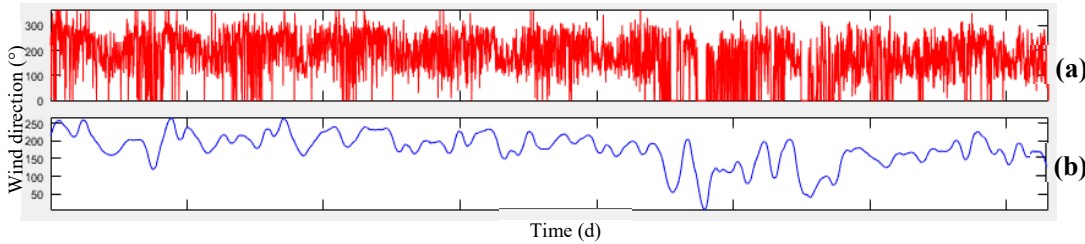

**Figure 17.** Time series (**a**) unrefined (**b**) refined wind direction in Chitgar Station.

As shown in Figures 12–17, Savitzky-Golay filter has been able to delete the irregular wavelets and abrupt changes; however, there are still data deficiencies in the signals because where the data is unavailable, Savitzky-Golay filter selects the zero signals as default. For this purpose, Fourier series and spline functions are used to compensate for the missed data. Figures 18 and 19 compare the time series Fourier series and spline functions of minimum temperature.

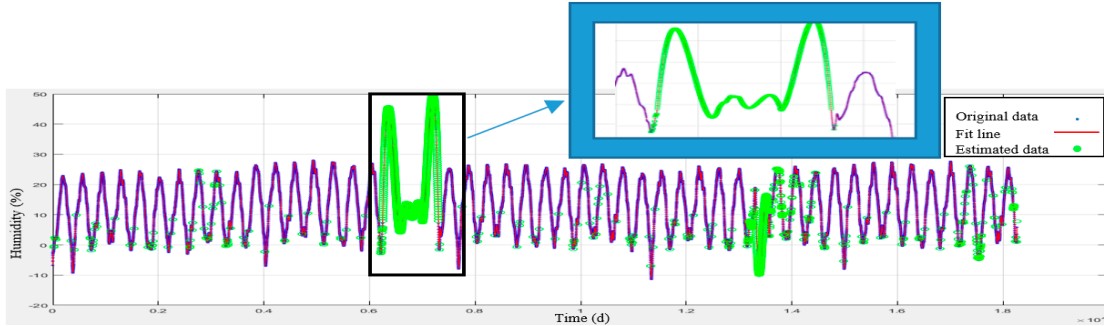

**Figure 18.** Compensation for missing data by fitting the spline function.

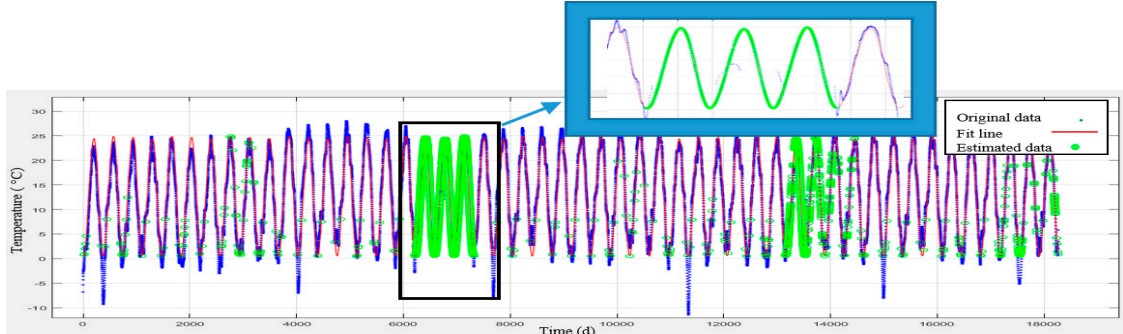

**Figure 19.** Compensation for missing data by fitting the Fourier series function.

Given that the spline function locally corresponds with the points, for long times of the missed data, it cannot appropriately estimate the monthly alternations but for the short time data missing (daily or weekly), it can appropriately estimate the missed data even much better than Fourier series. As shown in Figure 19, the Fourier series has managed to maintain and estimate the period and data alternation. To this end, this research uses spline for daily and weekly data loss and Fourier series for monthly data loss. The correlation coefficient ($R^2$) of the spline method was 98% and that of Fourier series was 83%.

Two datasets were employed for prediction of air pollution in the city of Tehran, which differ in terms of refined and non-refined data for the air pollutants. For this reason, a Savitzky-Golay filter was used in order to delete the noise and the irregularities in the time series of the pollutants. In this regard, the effects of a Savitzky-Golay filter on the error of the pollutants prediction are presented in Figures 20 and 21.

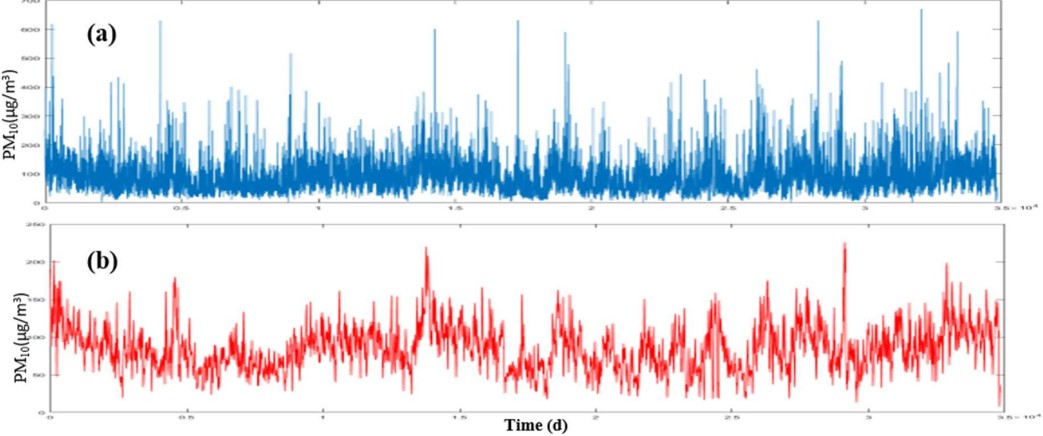

**Figure 20.** Savitzky–Golay filter related to the $PM_{10}$ pollutant (**a**) unrefined (**b**) refined time series.

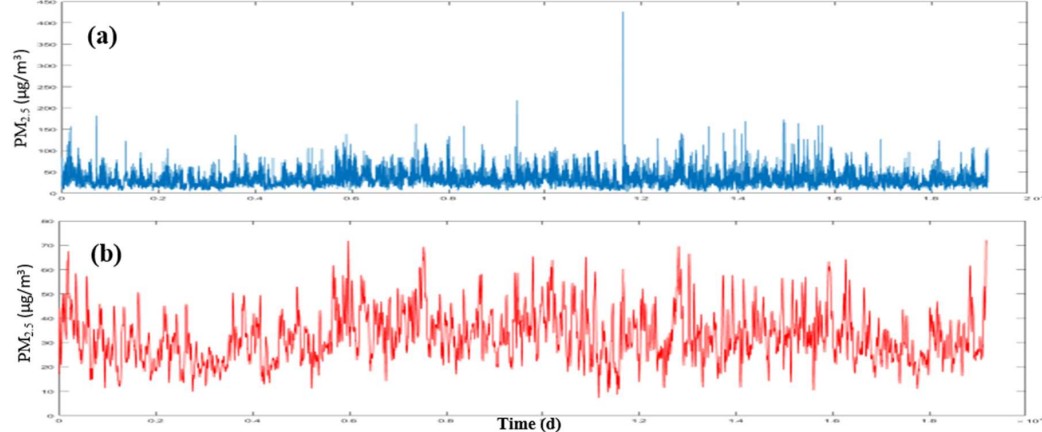

**Figure 21.** Savitzky–Golay filter the PM$_{2.5}$ pollutant (**a**) unrefined (**b**) refined time series.

As it clearly appears in Figures 20 and 21, this filter has appropriately managed to remove the irregularities and abrupt changes while keeping the data structure unchanged.

To develop a suitable structure for data to run the algorithm it is necessary to know the values of all the parameters at all the stations for the air pollution. Therefore, interpolation is required to spatially transfer the information from the meteorological stations to the air pollution stations. Considering that there is a limited number of meteorological stations for interpolation (five stations), IDW has performed as the most qualified interpolation method compared to those of Kriging and polynomial interpolations. Hence, IDW interpolation has been employed in this research. The results from the interpolation of parameters are presented in Table 1 where RMSE is calculated according to Equation (2).

**Table 1.** The results of meteorological parameters interpolation.

| Meteorological Parameter | RMSE |
|---|---|
| Wind speed (m/s) | 1.20 |
| Wind direction (°) | 14.6 |
| Minimum temperature (°C) | 1.1 |
| Maximum temperature (°C) | 2.8 |
| Humidity (%) | 5.16 |
| Air pressure (pa) | 19.04 |

### 3.2. Air Pollution Prediction

For the prediction of air pollution, the data need to be spatio-temporally harmonized. The structure of the input and output data for SVR, geographically weighted regression (GWR) and artificial neural network are presented in Table 2 and data structure in NARX neural network are presented in Table 3.

**Table 2.** The structure of the input and output data for support vector regression (SVR), geographically weighted regression (GWR) and artificial neural network (ANN).

| Inputx (t-d) | | | | | | | | | | | | Output |
|---|---|---|---|---|---|---|---|---|---|---|---|---|
| $D_{t-d}$ | $M_{t-d}$ | X | Y | Z | $WS_{t-d}$ | $DS_{t-d}$ | MxTt-d | $MnT_{t-d}$ | $H_{t-d}$ | $P_{t-d}$ | $KNN_{t-d}$ | $\dfrac{PM_{10}^{t}}{PM_{2.5}^{t}}$ |

**Table 3.** The structure of the input and output data for nonlinear autoregressive exogenous (NARX) neural network.

| Input | | | | | | | | | | | | Output |
|---|---|---|---|---|---|---|---|---|---|---|---|---|
| x(t-d) | | | | | | | | | | | y(t-d) | |
| $D_{t\text{-}d}$  $M_{t\text{-}d}$  X  Y  Z  $WS_{t\text{-}d}$  $DS_{t\text{-}d}$  $MaxT_{t\text{-}d}$  $MinT_{t\text{-}d}$  $H_{t\text{-}d}$  $P_{t\text{-}d}$  $KNN_{t\text{-}d}$ | | | | | | | | | | | $PM_{10}^{t-d}$ $PM_{2.5}^{t-d}$ | $PM_{10}^{t}$ $PM_{2.5}^{t}$ |

A separate prediction is conducted for each pollutant ($PM_{10}$ and $PM_{2.5}$). $D_{t-d}$, $M_{t-d}$, X, Y, $WS_{t-d}$, $DS_{t-d}$, $MxT_{t-d}$, $H_{t-d}$, $P_{t-d}$ and $KNN_{t-d}$ are day of week, month of year, the components of spatial coordinates, wind speed, wind direction, maximum temperature, minimum temperature, air humidity and air pressure, respectively. k is the nearest neighbor related to d days before predicting the value of the pollutants $PM_{2.5}^{t}$ and $PM_{10}^{t}$ on the prediction day.

Considering the difference between the data units, it is needed that all data are normalized using Equation (1).

$$n'_{ij} = \frac{(n_{ij} - min_j)}{(max_j - min_j)} j = 1, 2, \ldots, 9 , i = 1, 2, \ldots, k \tag{3}$$

In Equation (3), $n_{ij}$ is the relevant data for row i and parameter j, $n'_{ij}$ is the modified data for the scale space [01], $min_j$ and $max_j$ are the minimum and maximum parameters j, respectively.

The results of the prediction algorithms for the first dataset (pollutants have not been refined) and the second dataset (pollutants have been refined by Savitzky-Golay filter) for one day delay or predict have been presented in Tables 4 and 5.

**Table 4.** The results of the first dataset prediction algorithms.

| Algorithm | $PM_{10}$ | | | $PM_{2.5}$ | | |
|---|---|---|---|---|---|---|
| | RMSE ($\mu g/m^3$) | $R^2$ | Processing Time (s) | RMSE ($\mu g/m^3$) | $R^2$ | Processing Time (s) |
| SVR | 36.87 | 0.36 | 251 | 13.94 | 0.36 | 225 |
| GWR | 39.14 | 0.21 | 121 | 16.17 | 0.23 | 81 |
| ANN | 40.84 | 0.52 | 8 | 18.46 | 0.53 | 11 |
| NARX | 33.68 | 0.7 | 17 | 12.61 | 0.68 | 15 |

**Table 5.** The results of the second dataset prediction algorithms.

| Algorithm | $PM_{10}$ | | | $PM_{2.5}$ | | |
|---|---|---|---|---|---|---|
| | RMSE ($\mu g/m^3$) | $R^2$ | Processing Time (s) | RMSE ($\mu g/m^3$) | $R^2$ | Processing Time (s) |
| SVR | 17.01 | 0.68 | 1183 | 5.75 | 0.70 | 263 |
| GWR | 20.56 | 0.33 | 104 | 6.89 | 0.56 | 65 |
| ANN | 21.31 | 0.70 | 26 | 7.18 | 0.72 | 9 |
| NARX | 1.79 | 0.99 | 14 | 0.72 | 0.99 | 17 |

As shown in Table 4, the NARX method for both of the pollutants in the first datasets gives the minimum error and the best determination coefficient, followed by SVR with better performance than GWR, followed by ANN. In addition, for the second dataset and for both of the pollutants, NARX method leads to the minimum time calculation, minimum error, and the best determination coefficient, followed by SVR in terms of error index that performed better than GWR, followed by ANN.

In Tables 4 and 5, different algorithms were compared, in which the NARX datasets lead to the best results. In Table 6, the results from the second dataset were compared to that of the first dataset for all the methods.

**Table 6.** Comparison of the results of the second data set with those of the first data set.

| Algorithm | PM$_{10}$ | | | PM$_{2.5}$ | | |
|---|---|---|---|---|---|---|
| | RMSE (%) | R$^2$ (%) | Processing Time (%) | RMSE (%) | R$^2$ (%) | Processing Time (%) |
| SVR | −53 | +88 | −5 | −58 | +94 | +16 |
| GWR | −47 | +57 | −14 | −57 | +243 | −19 |
| ANN | −47 | +34 | +225 | −61 | +35 | −18 |
| NARX | −94 | +41 | −17 | −94 | +45 | +13 |

In Table 6, the positive and negative signs indicate the increase and decrease of the index expressed in percentages which have been calculated using Equation (4).

$$\text{Percentage change Index} = \left( \frac{\text{the value of index in the first dataset} - \text{the value of index in the second dataset}}{\text{the value of index in the second dataset}} \right) * 100 \qquad (4)$$

For example, the second dataset in the NARX method for both of the pollutants have reduced the RMSE error by 94%. The superiority of the second dataset relative to the first one is observed for all the methods. As a result, it can be mentioned that by removing the noise of pollutants time series, the accuracy of the prediction can be significantly enhanced which is more evident in the NARX method. Figures 22 and 23 illustrate the comparison of the observations and prediction maps of the afore-mentioned methods for PM$_{10}$ and PM$_{2.5}$ pollutants. The upper and lower limits of the air pollutions illustrated in Figures 22 and 23 are the maximum and minimum concentration of the air pollutions at the same day of the prediction.

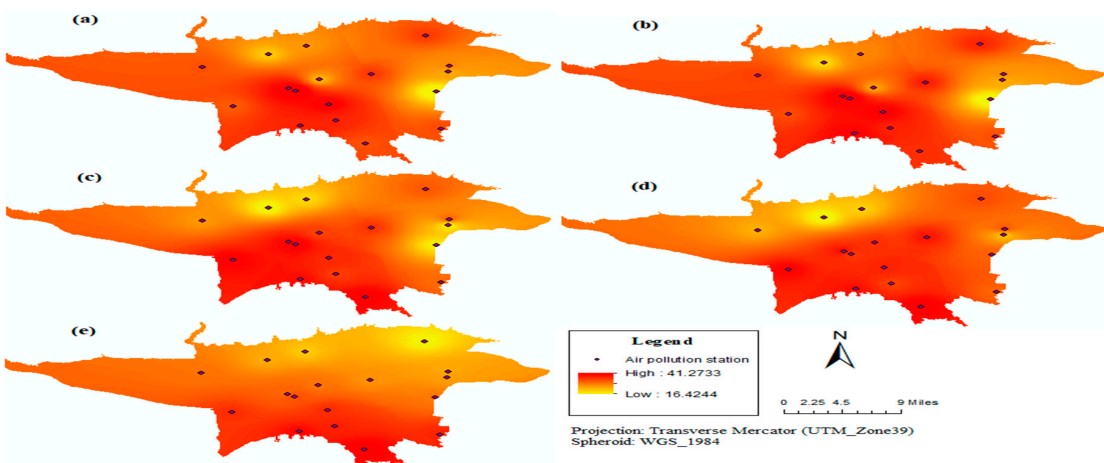

**Figure 22.** Comparison of (**a**) observation map and prediction maps of (**b**) NARX, (**c**) SVR, (**d**) GWR and (**e**) ANN methods for PM$_{2.5}$ (μg/m$^3$).

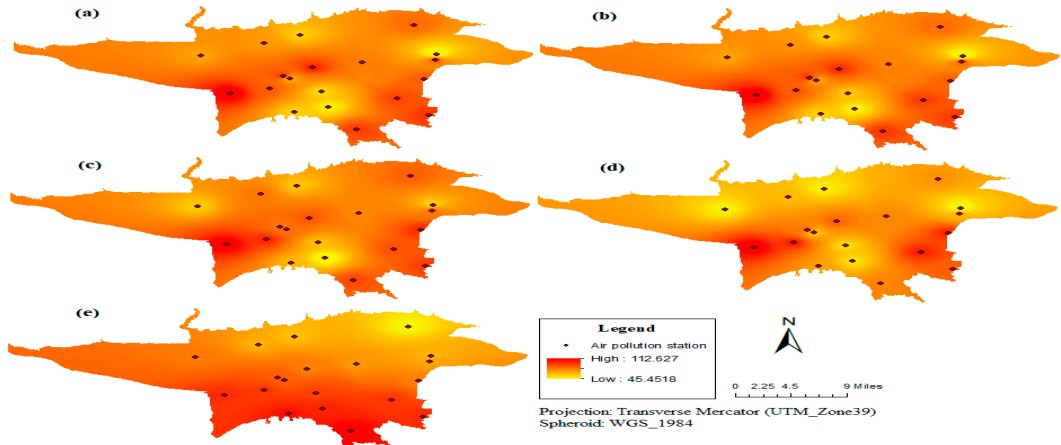

**Figure 23.** Comparison of (**a**) observation map and prediction maps of (**b**) NARX, (**c**) SVR, (**d**) GWR, and (**e**) ANN methods $PM_{10}$ ($\mu g/m^3$).

As shown in Figures 22 and 23, the greatest share of Tehran air pollution is concentrated in the southern part of the city because, according to Figure 2, wind direction in Tehran is often from west to the south and south-east of the city, which causes most of the pollution accumulated in the south of the city. In addition, the topography of Tehran is such that its north is at higher altitude and the south is almost flat. On the other hand, the south of Tehran is heavily populated and the buildings density is high, therefore, a higher air pollution accumulation is experienced in the south of the city compared to that of the north

According to Figures 22 and 23, and Table 5, the best and the most accurate method for prediction of the air pollution in the city of Tehran is the NARX method with the refined data. Table 7 illustrates the statistical properties of the error in NARX method for $PM_{10}$ and $PM_{2.5}$ contaminants. The best and worst points in Table 7 are related to the minimum and maximum errors, the magnitude of the mean absolute error (MAE), RMSE, the percentage of data whose error is less or more than the RMSE.

**Table 7.** The statistical characteristics of errors in the NARX method for $PM_{10}$ and $PM_{2.5}$ contaminants.

| Pollutant | Best Point ($\mu g/m^3$) | Worst Point ($\mu g/m^3$) | MAE ($\mu g/m^3$) | RMSE ($\mu g/m^3$) | Variance of Errors ($\mu g/m^3$) | <= RMSE (%) | >RMSE (%) |
|---|---|---|---|---|---|---|---|
| $PM_{2.5}$ | $1.5 \times 10^{-5}$ | 8.06 | 0.58 | 0.72 | 0.08 | 75.7 | 24.3 |
| $PM_{10}$ | $5.6 \times 10^{-5}$ | 12.24 | 1.45 | 1.79 | 0.59 | 77.6 | 23.4 |

The higher delay means the longer time is available to respond to the air pollution. For this purpose, to determine the highest delay in the cases where the prediction accuracy is significant, different models for NARX delay have been implemented for the second dataset, as shown in Table 8.

**Table 8.** Different time delay (days) modes in the NARX method for the second data set.

| Delay | $PM_{10}$ | | $PM_{2.5}$ | |
|---|---|---|---|---|
| | RMSE ($\mu g/m^3$) | $R^2$ | RMSE ($\mu g/m^3$) | $R^2$ |
| 1 | 1.79 | 0.997 | 0.72 | 0.997 |
| 2 | 3.28 | 0.993 | 1.32 | 0.991 |
| 3 | 4.82 | 0.986 | 1.96 | 0.982 |
| 4 | 6.19 | 0.978 | 2.50 | 0.97 |
| 5 | 7.43 | 0.969 | 3.03 | 0.957 |
| 6 | 8.79 | 0.956 | 3.59 | 0.938 |
| 7 | 9.98 | 0.94 | 4.06 | 0.92 |

As indicated in Table 8, with an increase in delay, the error increases and the determination coefficient is decreased. So it can be mentioned that for prediction of pollutant $PM_{10}$ in the next 7 days, the proposed method of this research will have a significant RMSE of 9.98 ($\mu g/m^3$). Considering the previous studies such as [18], in this study, the wavelet analysis was used for deletion of noise of the air pollutants and its results were compared to those of Savitzky–Golay filter as shown in Table 9.

**Table 9.** Comparison of the wavelet analysis and Savitzky–Golay filter results on prediction of the pollutants.

| Noise Removal Method | $PM_{10}$ | | $PM_{2.5}$ | |
|---|---|---|---|---|
| | RMSE ($\mu g/m^3$) | $R^2$ | RMSE ($\mu g/m^3$) | $R^2$ |
| Wavelet analysis | 6.73 | 0.941 | 5.16 | 0.953 |
| Savitzky-Golay | 1.79 | 0.997 | 0.72 | 0.997 |

As indicated in Table 9, the superiority of the Savitzky–Golay filter relative to that of wavelet analysis for both indices for determination of the correlation coefficient and error are verified. As a result, this filter can be used for the prediction of air pollution. One of the other objectives of the present research is determining the effective parameters in the prediction of $PM_{10}$ and $PM_{2.5}$ pollutants. For this purpose, this research implemented a genetic algorithm to determine the set of parameters which have the minimum error to predict the air pollution. The chromosomes of this genetic algorithm are 10-dimensional, representing 10 parameters for prediction of the air pollution. The values of each of dimension are binary, so that the value of one or zero for each of these, respectively indicates the presence or absence of the dimension (parameter) to consider the effective parameter in the prediction algorithms. Table 10 presents the effective parameters in each prediction method specified by the genetic algorithm. The value of 1 indicates the effectiveness of that parameter in the accuracy of the prediction method and the value of 0 means that the parameter is not effective in predicting the accuracy of the method.

**Table 10.** Determination of effective parameters by genetic algorithm for prediction of the air pollution.

| Method | Week Day | Month Year | Topography | Wind Speed | Wind Direction | Minimum Temperature | Maximum Temperature | Humidity | Air Pressure | Two of the Nearest Neighbors |
|---|---|---|---|---|---|---|---|---|---|---|
| NARX | 1 | 1 | 1 | 1 | 1 | 0 | 1 | 1 | 0 | 1 |
| ANN | 1 | 1 | 1 | 0 | 1 | 1 | 1 | 1 | 0 | 1 |
| GWR | 1 | 1 | 1 | 0 | 1 | 1 | 1 | 0 | 1 | 1 |
| SVR | 1 | 1 | 1 | 0 | 1 | 0 | 1 | 1 | 0 | 1 |

According to Table 10, wind speed predictions only NARX was applicable, while minimum temperatures were found predictable only with ANN and GWR and air pressure predictions only GWR was applicable. the parameters of day of week, month of year, wind direction, maximum temperature, and the values of pollutants for the two nearest neighbors, are the most effective parameters for prediction of air pollution in the city of Tehran.

## 4. Discussion

Selection of the statistical method to predict air pollution was one of most important objectives of this research. Unlike other methods for prediction of air pollution (i.e., chemical and physical), it is verified that the statistical method implemented has its low-volume of calculations, lower cost and presents an acceptable accuracy. Also, the capability of data interpretation and description is another advantage of the employed statistical method where statistics and geospatial information systems (GIS) intersect.

Given the important and applicable factors of the statistical method, in this case, one drawback of the statistical method, particularly machine learning, is that this technique is in the form of a black box. Therefore, it is impossible to individually analyze the precise amount of the effect of each parameter.

Of course, the subject itself will significantly help the prediction; that is, the effect of each parameter may be recognized individually in physical and mathematical relations, while the effect of other parameters, which do not exist in the relation, may be ignored. However, by using machine learning technique it is possible to discover a hidden pattern that exists among the data.

## 5. Conclusions

Measurement errors have been an integrated part of observations which cause errors in the modeling and analysis processes. Similarly, the finding of this research was not free from errors. Therefore, after investigating the various methods for noise and error removal, the Savitzky–Golay filter was found to be more appropriate than other methods such as wavelet analysis. This filter cannot alone compensate for the lost data. Therefore, this research uses Fourier series and spline functions, the former being appropriate for daily and weekly lost data and the latter for the monthly lost data. The extent of the effects of these methods on the accuracy improvement of the prediction model was nearly 10–15%. The two datasets were used in this research whose difference was in refinement or non-refinement of the times series of the pollutants. In both cases, the best method for prediction of air pollution was NARX method. When the time series of the pollutants is refined, about 40% increase in $R^2$ and 94% decrease in RMSE occurs. Considering the prior studies, this research had a minimum error in the air pollution prediction model.

The maximum prediction times were also addressed, which achieved a significant accuracy within 7 days. The results of this research are only for $PM_{10}$ and $PM_{2.5}$ pollutants as the main source of air pollution in Tehran. The applicable data were also collected from stations for air pollution measurement on the daily basis. Finally, using the genetic algorithm, the effective parameters in prediction of the air pollution, data for day of week, week of month, topography, wind direction, maximum temperature, and the value of pollutants for the two selected nearest neighbor air pollution stations were identified.

The major contributions of this research are as follows:

- A comparative study of machine learning methods including NARX, ANN, GWR and SVR has been employed for air pollution prediction and the NARX finally selected as the optimum one.
- The research has improved the efficiency of the machine learning method employed based on filtering the existing noise in both the meteorological and air pollution data as well as predicting the missed meteorological data.
- This research has proposed a novel approach for air pollution prediction in urban areas based on both stationary and non-stationary pollution sources using machine learning and statistical methods.
- Fourier series has been implemented for monthly missed meteorological data and spline methods employed for daily and weekly missed meteorological data.
- The effective parameters for air pollution prediction have been determined in this research.

Some shortcomings of the research are as follows:

- Unviability of some urban traffic data, distance from road and some more meteorological parameters related to pollution sources causes some errors in the final air pollution prediction.
- As some of the air pollution data provided by the Iranian Environmental Protection Agency were heterogeneous and lack a suitable spatio-temporal structure, they have been excluded in the analysis.
- The employment of Savitzky–Golay filter in some limited places where no air pollution data were available, may slightly affect the final air pollution prediction accuracy which could not be avoided due to some data scarcity. Anyway, the implementation of Fourier series and spline functions in those areas has significantly reduced the impact of the data gap.

For future research, the following recommendations are presented to enhance the quality of air pollution prediction:

- This research used daily data of the pollutants. So, the quality of the proposed model can be significantly improved if hourly data is implemented
- Considering the importance of air pollution problem, it is recommended that the number of air pollution measuring stations increases in Tehran so as to allow for a better fit on the air pollution prediction.
- Applying more inclusive parameters such as urban traffic, distance from road and some more meteorological parameters related to pollution sources are among the important factors in modeling and prediction of urban air pollution especially in industrial cities like Tehran. In this research, due to the unavailability of the afore-mentioned data, it was impossible to apply all the influencing data in the proposed model.

**Author Contributions:** Conceptualization: M.R.D., K.F. and A.G.; Data curation: A.G.; Formal analysis: A.G. and M.R.D.; Funding acquisition: M.R.D.; Methodology: M.R.D. and A.G.; Project administration: M.R.D.; Resources: S.H.A.; Validation: M.R.D. and A.G.; Visualization: A.G.; Writing—original draft: A.G. and M.R.D.; Writing—review & editing: M.R.D., G.R.S., Y.R. and G.R.N.

**Funding:** This research was funded by the Ministry of Science, Research and Technology, Tehran, Iran, grant No. 5309340 allocated to the first author.

**Acknowledgments:** The authors would like to acknowledge AQCC and the Iran Environmental Protection Agency for providing air pollution data, Iranian Meteorological Organization for providing meteorological data and Christophe Claramunt for his critically reviewing the paper. This work has been Supported by the Center for International Scientific Studies & Collaboration (CISSC), Ministry of Science, Research and Technology, Iran.

**Conflicts of Interest:** The authors declare no conflict of interest.

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
