# Peer review of "A Novel Method for Improving Air Pollution Prediction Based on Machine Learning Approaches: A Case Study Applied to the Capital City of Tehran"

_ijgi, doi:10.3390/ijgi8020099_

Round 1

Reviewer 1 Report

The study focusses on particulate concentrations in the
metropolitan area of Tehran often charged with serious air
pollution like smog. Thus predicting near-future pollution levels
is important for the public in order to adapt and minimize
suffering. The authors use state owned representative air
control measuring stations and meteorological stations to
compare different methods for interpolation and for nowcasting
of pollution levels at low budget. While of notable interest,
some aspects remain speculative and should be laid out clearly
and followed: Note the criteria the predictions should meet to
serve as warnings in case of high levels and for administrators
to decide for taking measures to counteract. A comparison of
methods and avaiable data is not accessible. At which point the
method works best, at which range an applicant needs to be
moderately suspicious (limits of approach)? The study is
certainly of interested and meets the scope of the journal.
But some work is definitely needed before final acceptance.

General comments:
*LANGUAGE: Check language and be more concrete: "her
administrators" -> "its administrators and residents".
Are administrators living elsewhere? "... have long
been struggling" -> with what? Measures? Health issues?
*A considerable proportion of air pollution..is
attributed to PM10 and PM2.5" by which studies, instruments, approaches?
Suggestion, reformulate such as "a serious health thread is particulate
pollution. Therefore, the present study..."
*"To model the air pollution prediction", prediction is
a model! Use different wording such as "In order to develop
an accurate prediction algorithm ..."
*General: Assume readers not to have taken part in
your study but introduce them to it along a clear distinct path,
easy to grasp. Words such as 1 day prediction error of 1.79
microg/m3 need to be compared to representative values.
Is this huge for Tehran conditions or not? Relative is allways
preferrable and a range makes infos even more reliable.

# Introduction
* It's allways welcome to clearly determine terms of major importance
in this investigation such as 'air pollution', because a wide scatter of different
associations is linked with including toxic gases, particles
with and without humidity (smog). A series of different approaches
to classify current air pollution and to make predictions for the
next hours has already been developed elsewhere, which may require
initial investment but can be kept at a moderate level. However,
this allows urban administrators to find best measures to counteract
the consequences and to fix guidelines to prevent.
* Provide more references for lines 53-59. A single and then this one
is hardly sufficient in this complex topic. The better the statements
- including references - the more thrustworthy they become.
l.59/60: Again, be more concrete! "One of the objectives is to provide
citizens with information to make them aware of air quality rate"
could is very vague. How, by web, social media, development clips
for the next hours frequently updated? Official website? And what
is 'air quality rate'? Do you mean ranking? A rate is a change
per time!
*Define the difference between 'model' and 'prediction'! I assume
both identical, except that a model covers a larger time range i.e.
simulations of past, present and future conditions. Please use a
standard science book for reference in here (l.60ff)
*l.64ff: Please define 'air pollution' accurately. ..."PM10 and PM2.5
constitute the highest share of air pollution"? Compared by what and
to what? By volume or mass? Only particles or NOx as well? What about
CO and further toxic gases such as SO2? Compared to particle mass
that's evident since a single 10 micrometer in diameter particle
consisting of dust or sand weights >1000 times a 1 micrometer
in diameter particle. But with respect to health issues the smaller ones
are much more dangerous because of their penetration into the veins
and the transport of toxic substances throughout the body.
* While the authors approach is worth conducting, I am sure there
are more tools which could be used for emission data access, e.g.
satellite data, emission inventories and a step by step improval of
predictions minimizing the prediction errors in altering local
emissions. Please argue, why this approach has been favoured, costs?
Because input of measurement stations requires continuous calibration,
which is not necesarily performed causing errors in the calculations
to follow.
* Consider statistical models as tools but not as entire wisdom.
Such models are developed for certain conditions. If applied elsewhere
this may lead to a blackbox running machine with a difficulty to interpret!
l.97: Please explain "the prediction precision of PM2.5 to the average of 90"
in a more detailed manner to the reader, especially when discussing RMSE of
of .76 etc beforehand. Stated in percent?

# Methods
* A lot of different stistical methods have been tested/applied to the
about 10 years dataset from Tehran. A worthy and time consuming step.
Some aspects remain still unclear to me:
a) you applied the Savitzky-Golay-filter for smoothing the data. What
was the uppermost potential (filter order) used? Which effects did you
ignore by doing that?
b) Based on which criteria the choice of best model was made,
i.e. minimum least square error, based on
particulates only or gases too?
c) Was a seasonality effect observed, e.g. a better match for a certain
approach in wintertime for one but for a different during
summertime?
Please note the different histograms of parameters such as
temperature (normally distr.), O3, normal or binormal, PM (event type
character). This may cause a different match since both, i.e.
measurements and simulations both can and will produce errors depending
on exactness of measurement conditions and behaviour description.
While Figure 1 is of interest to the reader a list of clear focus,
criteria to meet and steps followed is recommended. The same goes
for the data used: Meteorological and air pollution data are two big
terms, which may contain a wide range of variables. Please provide a
table or list.
* Which timeframe has been used? Which stations are addressed? You
may put this into supporting online information, but makes your analysis
more transparent. Otherwise one may think as tuned and selected as
long as all mismatching datapoints are excluded.

# Results
l.184: It's nice to know about every fifth Iranian is living
in the metropolitan area of Tehran. But state the number of inhabitants.
This would give a much better idea about emissions by transport
vehicles, wood burn etc.
* One important question for me ist the relative contribution of
internal and external emissions for air pollution in Tehran.
What is the average contribution of within city boundaries created
particulates and what about dust storms? It would even highlight
the need for an accurate prediction if local sources are mostly predominant.
Otherwise errors may be smoothed out.
* Question: can you provide a Google map or whatever geographical figure
to indicate locations of geography, air poll. and meteo measurement sites?
This would make the authors challenge even more evident for anyone interested
in ("Sometimes a picture can tell more than 1000 paragraphs.").
* Provide any reference or location of the Chitgar measurement site used
for Figs. 3-8 and 14. There is no detailed word about it in the text.
* A windrose plot such as of openair toolpackage of R would be a good
way to indicate conditions. Otherwise wind directions and speed are
difficult to connect. PLEASE change the time axis to years! That would
allow an idea about seasonaility and its drawbacks. PROVIDE coefficients
of smoothing either as table in the study or as supp. online info.
* There is evidently a problem of your data in Figs. 11+12, i.e. data gaps.
Time gaps should be excluded, but not smoothened by a filter! This does
not preproduce missing data but make the gap more flat! In order to
"regain" the data witha certain probability, you need to set up functional
relations to accessible data (correlation, regression etc.). The smoothing
works only for one or two ming data points depending on your filter order.
Therefore, the presented results for that particular timeframe including
weeks of missing data do not supply
trustworthy data used in the following! Leave blacks at missing
data or use further methods e.g. correlations with nearby stations etc.
Which analysis program has been used?
Fig. 14 indicates several challenges: (a) A minimum required wind
velocity of about 2 m/s that changes after about 60% of the time! Most
likely this is due to different instruments. Those should not be used
lumped as a single dataset, although the intension is clear. If the
histogram is plotted a binormal distribution should be seen for the
first 60% of the data with zeros and data beyond 2 m/s, and a continuous
plot for the latter, indicating that suddenly smaller values were present
that formerly weren't. Suggestion: Skip all values of 0 or change them to
NA. Test you results by splitting the dataset into two parts and perform
the analysis for each of them. Is there a significant change? I am sure
parts of RMSE in Table 1 refer to this issue.
* Predictions rely on good input data and description of processes. Why not
stating: "NO prediction for 10 am since measurement xy was missing that time"?
Otherwise you create artificial data. If you still stick to those provide
uncertainty ranges.
*Fig. 17+18: Very nice indeed. Especially the ones with the Savitzky-Golay
filtered data. Can you retrieve the traffic contribution to this? Rush hour
peaks show up nicely, i.e. higher in the morning with smaller boundary layer
height (BLH) but noticeble in the afternoon/evening at maximum (BLH).
* Table 4 and l.274ff: There is an evident mismatch in the interpretation of
RMSE and R^2. While ANN does provide a much better R^2 (i.e. better fit) for
both PM10 and PM2.5, the RMSE is notably worse! I suspect this to originate
from a set of outliers the model could hardly cope with (e.g. very high values)
that drove the RMSE to very high levels. But it would be essential to figure
this out since an R^2 of .36 for SVR and 0.52 for ANN is clearly pointing
towards ANN to be the better one.
* Table 5: An R^2 is never >1! I guess the '68' in the first line for
SVR refers to 0.68! Correct it.
* Personally I have some problems with Equation (2), because of its
relative character. It may improve relatively to a huge extend but
lead to awfull results anyhow. The NARX still is the best. I would
propose taking an absolute reference as "optimum" to compare with.
Using that it could be stated "by the second dataset the mismatch of
e.g. SVR was improved substantially to an R^2 of...".
*Figs. 19 and 20: Coming back to Tables 4 and 5 and the match of models.
Fig. 19 clearly indicates the horizontal difference on I guess hotspots, which
differs for the individual methods. SVR is basically dividing Tehran into
two halves, while NARX highlightens hotspots and ANN and GWR are somewhat the
interpolation between NARX and SVR, reflected in the R^2 values. Therefore
I conclude with the focus of the total area NARX operates best followed by ANN,
GWR and SVR.
* Table 9: Right part is missing. The worth of Table 9 is not obvious to me.
A paragraph discussing the applicability of methods such as "for wind speed
predictions only NARX was applicable, while minimum temperatures were found
predictable only with ANN and GWR" seems more usefull. Skip the table.
* l.318: Remove brackets around 9.

Author Response

Response to Reviewer 1 Comments

Point 1: Note the criteria the predictions should meet to  serve as warnings in case of high levels

Response 1: line370 and 371 (in new verson) have been added:

The upper and lower limits of the air pollutions illustrated in Figures 22 and 23 are the maximum and minimum concentration of the air pollutions at the same day of the prediction.”

Point 2: and for administrators  to decide for taking measures to counteract

Response 2: lines 62 to 64 have been added:

“ In addition, the concerned city managers can implement the information to control the urban traffic and the responsible pollutant industries and to increase public transport facilities in order to mitigate the level of the pollution.”

Point3 :A comparison of methods and available data is not accessible

Response 3:

Available data:

 Lines 153 to 155 have been added:

In general, two types of methods including satellite imagery and ground sensors are used to collect air pollution data. Given the cost, availability and accuracy of ground sensors data for 10 years, this type of data has been used in this research.”

A comparison of methods:

In the lines 93 to 97, the methods were compared:

In the past few decades, two general approaches of deterministic and stochastic methods have been used to predict air-pollution [11]. Diffusion models are among the deterministic methods developed in various regions for modeling and monitoring the air pollution [12], [13]. However, the output of these models relies on the input data, and in order to use them, it is necessary to access the data on how the pollutants disseminate and diffuse in atmosphere [14].”

Point 4: At which point the method works best

Response 4: in lines 383 to 386 and table 7 have been added.

Point 5: at which range an applicant needs to be moderately suspicious (limits of approach)?

Response 5: Lines 469 to 478 have been added:

Some shortcomings of the research are as follows:

·            Unviability of some urban traffic data, distance from road and some more meteorological parameters related to pollution sources causes some errors in the final air pollution prediction.

·            As some of the air pollution data provided by Iranian Environmental Protection Agency were heterogeneous and lack a suitable spatio-temporal structure, they have been excluded in the analysis.

·            The employment of Savitzky-Golay filter in some limited places where no air pollution data were available, may slightly affect the final air pollution prediction accuracy which could not be avoided due to some data scarcity. Anyway, the implementation of Fourier series and spline functions in those areas has significantly reduced the impact of data gap.

Point 6: "... have long been struggling" -> with what? Measures? Health issues?

Response 6: lines 25 and 26 have been added:

“with air pollution damage such as the health issues of citizens”

Point 7: A considerable proportion of air pollution..is attributed to PM10 and PM2.5" by which studies, instruments, approaches? 

Response 7: Lines 66 to 70 have been corrected:

“According to the latest available statistics from 21 stations belonging to Tehran Air Quality Control Company (AQCC) and 16 air-pollution measurement stations belonging to the Iranian Environmental Protection Agency, PM10 and PM2.5 constitute the highest proportion concentration of air-pollution in Tehran. Among the pollutants such as CO, O3, NO2, SO2, PM10 and PM2.5, PM2.5 has the highest share.”

Point 8: Words such as 1 day prediction error of 1.79  microg/m3 need to be compared to representative values. Is this huge for Tehran conditions or not? Relative is allways preferrable and a range makes infos even more reliable.

Response 8 : in lines 385 to 388 and table 7 have been added

Point 9: It's allways welcome to clearly determine terms of major importance in this investigation such as 'air pollution'.

Response 9: in lines 48 to 50 have been added:

“ Air pollution means the existence of one or more pollutant contaminated the outdoor or indoor air in various amounts and periods which may harm human, vegetation or animal life or unexpectedly interacts with normal life or properties [1], [2].”

Point 10:  A series of different approaches to classify current air pollution and to make predictions for the next hours has already been developed elsewhere, which may require initial investment but can be kept at a moderate level. However, this allows urban administrators to find best measures to counteract the consequences and to fix guidelines to prevent. 

Response 10: In lines 93 to 97, it is said:

In the past few decades, two general approaches of deterministic and stochastic methods have been used to predict air-pollution [11]. Diffusion models are among the deterministic methods developed in various regions for modeling and monitoring the air pollution [12], [13]. However, the output of these models relies on the input data, and in order to use them, it is necessary to access the data on how the pollutants disseminate and diffuse in atmosphere [14].”

Point 11: l.59/60: Again, be more concrete! "One of the objectives is to provide 
citizens with information to make them aware of air quality rate" 
could is very vague. How, by web, social media, development clips 
for the next hours frequently updated? Official website? And what 
is 'air quality rate'? Do you mean ranking? A rate is a change 
per time!

Response 11: this information includes the amount of daily changes in PM10 and PM2.5, which can informed to people and authorities through organizations, sites, and apps.

Point 12:  Please define 'air pollution' accurately. ..."PM10 and PM2.5
constitute the highest share of air pollution"? Compared by what and
to what? By volume or mass? Only particles or NOx as well? What about
CO and further toxic gases such as SO2? Compared to particle mass 
that's evident since a single 10 micrometer in diameter particle 
consisting of dust or sand weights >1000 times a 1 micrometer 
in diameter particle.

Response 12: Lines 68 and 70 have been corrected:

“PM10 and PM2.5 constitute the highest proportion concentration of air-pollution in Tehran. Among the pollutants such as CO, O3, NO2, SO2, PM10 and PM2.5, PM2.5 has the highest share.”

Point 13: But with respect to health issues the smaller ones 
are much more dangerous because of their penetration into the veins
and the transport of toxic substances throughout the body.

Response 13: lines 79 t0 92 have been added and corrected:

“PM2.5 contaminants contain particles that are created by combustion or caused by the formation and compression of secondary particles. PM10 particles contain particles that are 10 micrometer in diameter and smaller and can pass through the first defensive barrier (nose and throat), damage the lungs and deposition there[8]. Studies have shown that exposure to suspended particles is associated with health effects such as cardiovascular and respiratory diseases[9]. The World Health Organization estimates that if the average annual concentration of PM10 is reduced from 70 μg/m3 to 20 μg/m3, then the associated deaths will be reduced by 15%[10]. In fact, there is a relationship between the exposure to intense concentrations of suspended particles and the increase in daily and annual mortality, as well if the concentration of these pollutants is reduced if other factors are fixed; the associated deaths are reduced [10]. These particles are very tiny and their damage to human health is higher. In this study, these pollutants are used as pollutants for predicting air pollution. Hence, air-pollution prediction is becoming one of the managerial solutions to prevent and/or mitigate its destructive implications. Therefore, it seems necessary to predict PM10 and PM2.5 pollutants using the appropriate methods. “

Point 14: While the authors approach is worth conducting, I am sure there
are more tools which could be used for emission data access, e.g. 
satellite data, emission inventories and a step by step improval of 
predictions minimizing the prediction errors in altering local
emissions. Please argue, why this approach have been favoured, costs?
Because input of measurement stations requires continuous calibration,
which is not necesarily performed causing errors in the calculations
to follow.

Response 14: lines 153 to 159 have been added:

“In general, two types of methods including satellite imagery and ground sensors are used to collect air pollution data. Given the cost, availability and accuracy of ground sensors data for 10 years, this type of data has been used in this research. . One of the major problems of ground-based sensors is the calibration of the device. The air pollution data have been validated by Tehran Air Quality Control Company (AQCC). In spite of the above assumption, in this study, a data refinement mechanism was used to better estimate the concentration of pollutants in areas where there is a data gap or an error in the data.”

Point 15:  Consider statistical models as tools but not as entire wisdom. 
Such models are developed for certain conditions. If applied elsewhere
this may lead to a blackbox running machine with a difficulty to interpret!
l.97: Please explain "the prediction precision of PM2.5 to the average of 90"

Response 15: Lines 111 to 117 have been corrected:

[18] Introduced a model to improve the artificial neural network, which is a combination of air mass route analysis and wavelet transform. The rate of RMSE for the combinational model can be decreased by 40% in average. The study verified that especially in the days with higher concentration of PM2.5 often predicted for the warned threshold of the combinational models using wavelet analysis and detection rate (DR), the RMSE can reach to the average limit of 90%. This approach indicates the potential of the proposed model in air-quality prediction system in other countries.”

Point 16: in a more detailed manner to the reader, especially when discussing RMSE of
of .76 etc beforehand. Stated in percent?

Response 16: Lines 250 to 260 have been added.

Point 17: you applied the Savitzky-Golay-filter for smoothing the data. What 
was the uppermost potential (filter order) used? Which effects did you
ignore by doing that?

Response 17: Lines 292 and 294 hase been added:

The input parameters of the Savitzky-Golay filter include the polynomial order and frame length whose best fit for order = 3 and frame length = 23 was obtained.

Point 18: Based on which criteria the choice of best model was made, 
i.e. minimum least square error, based on 
particulates only or gases too? 

Response 18: minimum RMSE and maximum R2

Point 19: Was a seasonality effect observed, e.g. a better match for a certain 
approach in wintertime for one but for a different during 
summertime? 

Response 19: Lines 180 to 190 and figure 1 have been added

Point 20: Please note the different histograms of parameters such as
temperature (normally distr.), O3, normal or binormal, PM (event type 
character). This may cause a different match since both, i.e. 
measurements and simulations both can and will produce errors depending
on exactness of measurement conditions and behaviour description.

Response 21: Line 275 hase been added (they are observed value)

acquired from meteorological stations

Point 22: While Figure 1 is of interest to the reader a list of clear focus, 
criteria to meet and steps followed is recommended. The same goes 
for the data used

Response 22: in lines 229 to 249 it is explained

Point 23:  Meteorological and air pollution data are two big
terms, which may contain a wide range of variables. Please provide a 
table or list. 

Response 24: in table 2 and 3 and figure 3 have been shown.

Point 25: Which timeframe have been used? Which stations are addressed? You 
may put this into supporting online information, but makes your analysis
more transparent. Otherwise one may think as tuned and selected as 
long as all mismatching datapoints are excluded.

Response 25: in lines 212 and 213 and figure5 have been shown.

Point 26: l.184: It's nice to know about every fifth Iranian is living 
in the metropolitan area of Tehran. But state the number of inhabitants.

Response 27: in line 263 and 264 have been corrected.

That (20%) was  for Tehran province , not Tehran city.

with 10% of the country’s total population. In 2016, Tehran population was 8.6 million and Iran population was 81.1 million

Point 28: One important question for me ist the relative contribution of 
internal and external emissions for air pollution in Tehran. 
What is the average contribution of within city boundaries created 
particulates and what about dust storms? It would even highlight 
the need for an accurate prediction if local sources are mostly predominant. 
Otherwise errors may be smoothed out.

Response 28: Lines 70 to 78 have been added:

“Based on the studies undertaken in 2017 by AQCC and the technical report produced on the Tehran Air Pollution Prediction System, nearly 5% of PM2.5 pollutants are coming from neighboring populated areas laid in the west (city of Karaj, south west of Tehran (city of Shahryar), and south east of Tehran (city of Rey)). Such a percentage has been found higher in the summer time due to higher levels of wind speed in transporting the dust driven from out west and trapped in the Greater Tehran basin[7]. The percentage presented here on PM2.5 pollutant can be regarded, based on the AQCC expert opinion, as the highest percentage with respect to other pollutants that have been detected to be under 5%.  Furthermore, the PM2.5 detected in the winter time above is not of natural or wind blown dust from outside deserts[7].”

Point 29: can you provide a Google map or whatever geographical figure 
to indicate locations of geography, air poll. and meteo measurement sites?

Response 29: in figure 5 has been added.

Point 30: * Provide any reference or location of the Chitgar measurement site used
for Figs. 3-8 and 14. There is no detailed word about it in the text.

Response 31: it has been shown in figure 5.

Point 32:  Otherwise wind directions and speed are 
difficult to connect.

Response 32: in lines 191 to 198  and figure 2 have been added

Point 33: There is evidently a problem of your data in Figs. 11+12, i.e. data gaps. 
Time gaps should be excluded, but not smoothened by a filter! This does 
not preproduce missing data but make the gap more flat! In order to
"regain" the data witha certain probability, you need to set up functional
relations to accessible data (correlation, regression etc.). The smoothing 
works only for one or two ming data points depending on your filter order.

Response 33: in lines 287 to 294 have been explained and added.

Tehran meteorological and air pollution data have some missing values for PM10 and PM2.5 pollutants during the ten years period (2006 to 2016). Fourier series and splines have been used to remove the missed data for the meteorological data which is used as a training data in the machine learning process. Otherwise, less data would be available for the training of the machine learning process. In this research the selection of a period when both air pollution data and all the intended weather parameters were simultaneously available was necessary. The input parameters of the Savitzky-Golay filter include the polynomial order and frame length whose best fit for order = 3 and frame length = 23 was obtained.”

Point 34: Therefore, the presented results for that particular timeframe including
weeks of missing data do not supply 
trustworthy data used in the following! Leave blacks at missing 
data or use further methods e.g. correlations with nearby stations etc.
Which analysis program have been used?

Response 34: We used fourier series and spline  methods for missing data.

Point 35: Fig. 14 indicates several challenges: (a) A minimum required wind velocity of about 2 m/s that changes after about 60% of the time! Most 
likely this is due to different instruments. Those should not be used
lumped as a single dataset, although the intension is clear. If the
histogram is plotted a binormal distribution should be seen for the 
first 60% of the data with zeros and data beyond 2 m/s, and a continuous 
plot for the latter, indicating that suddenly smaller values were present
that formerly weren't. Suggestion: Skip all values of 0 or change them to 
NA. Test you results by splitting the dataset into two parts and perform
the analysis for each of them. Is there a significant change? I am sure 
parts of RMSE in Table 1 refer to this issue.

Response 35: While the claim of the respected reviewer regarding the change of wind speed presented in Figure 8 seems affects the results of the prediction model, the wind speed has not been finally selected in the optimization process using genetic algorithm. Therefore, wind speed has not been finally used as an defective parameter in the prediction model. In addition, the average wind speed has been considered in Figure 8, which may affect the final results slightly .

Point 36: Predictions rely on good input data and description of processes. Why not
stating: "NO prediction for 10 am since measurement xy was missing that time"?
Otherwise you create artificial data

Response 36: in lines 287 to 294 have been explained and added. We had to predict missing values for the reason below.

Tehran meteorological and air pollution data have some missing values for PM10 and PM2.5 pollutants during the ten years period (2006 to 2016). Fourier series and splines have been used to remove the missed data for the meteorological data which is used as a training data in the machine learning process. Otherwise, less data would be available for the training of the machine learning process. In this research the selection of a period when both air pollution data and all the intended weather parameters were simultaneously available was necessary. The input parameters of the Savitzky-Golay filter include the polynomial order and frame length whose best fit for order = 3 and frame length = 23 was obtained.”

Point 37: . If you still stick to those provide 
uncertainty ranges.

Response 37: lines 307 and 308 have been added:

The correlation coefficient (R2) of the spline method was 98% and that of Furious series was 83%.

Point 38: Fig. 17+18: Very nice indeed. Especially the ones with the Savitzky-Golay 
filtered data. Can you retrieve the traffic contribution to this?

Response 38: There is no traffic data in the city of Tehran for 10 years.

Point 39: Table 4 and l.274ff: There is an evident mismatch in the interpretation of 
RMSE and R^2. While ANN does provide a much better R^2 (i.e. better fit) for
both PM10 and PM2.5, the RMSE is notably worse! I suspect this to originate 
from a set of outliers the model could hardly cope with (e.g. very high values)
that drove the RMSE to very high levels. But it would be essential to figure
this out since an R^2 of .36 for SVR and 0.52 for ANN is clearly pointing 
towards ANN to be the better one.

Response 39: If R2 is high, this is not a reason for the lower RMSE

Point 40: Table 5: An R^2 is never >1! I guess the '68' in the first line for 
SVR refers to 0.68! Correct it.

Response 40: There was a mistake in writing and it have been corrected

Point 41:  Personally I have some problems with Equation (2), because of its 
relative character. It may improve relatively to a huge extend but 
lead to awfull results anyhow. The NARX still is the best. I would
propose taking an absolute reference as "optimum" to compare with.
Using that it could be stated "by the second dataset the mismatch of 
e.g. SVR was improved substantially to an R^2 of...".

Response 41: It is used to indicate the percentage of the index changes

Point 42: Figs. 19 and 20: Coming back to Tables 4 and 5 and the match of models. 
Fig. 19 clearly indicates the horizontal difference on I guess hotspots, which 
differs for the individual methods. SVR is basically dividing Tehran into 
two halves, while NARX highlightens hotspots and ANN and GWR are somewhat the 
interpolation between NARX and SVR, reflected in the R^2 values. Therefore 
I conclude with the focus of the total area NARX operates best followed by ANN, 
GWR and SVR.

Response 42: it's true. In Figure 19 and 20, ANN and SVR methods were written incorrectly. Has benn corrected.

Point 43: Table 9: Right part is missing. The worth of Table 9 is not obvious to me. 

Response 43:it has been corrected.

Point 44: A paragraph discussing the applicability of methods such as "for wind speed
predictions only NARX was applicable, while minimum temperatures were found
predictable only with ANN and GWR" seems more usefull. Skip the table.
* l.318: Remove brackets around 9.

Response 44: Done.

Reviewer 2 Report

Major comments:

The methodology used in this study were generally well-described, although they are not new to the audiences. The Introduction was too heavy (lines 45 -152 and account for almost 1/3 of content in the manuscript), and the worst part is after the long introduction, the value and contributions of this study couldn’t be identified clearly. Please consider revising section 1 thoroughly and highlight the objectives of this study with dot points at the end of section.

More structured and detailed and descriptions of the original 10-year measurements dataset (both air quality and Met data) used in this study are expected in section 2 “Material and Methods”. For instance,

There’s no mention to data QA/QC processes, did they have any QA/QC done before releasing to the users?

Are the minutes or hourly or daily data used in this study?

Is it possible to add a map showing the stations where the data are used?

There’s a well-designed flowchart shown in Figure 1, may I suggest the authors to follow this flowchart and make some dot points to organise all the information related to data acquisition, data refinement, interpolation, noise elimination etc. to help reader to follow.

Also, there are some data descriptions in the current section 3 (lines 205-213), I would suggest moving all the information related to data into section 2.

These are lots of interesting findings and discussions in the 3rd section “Results”, so may I suggest to authors to add at least two sub-sections to accommodate results for 1) data refinement, 2) prediction of air pollution.

Specific comments:

Lines 162-163, “these data are not error-free”, does this mean the data were not QA/QC before they can be released?

Line 172 “These filters are better than ordinary Finite Impulse Response”, any reference to support this?

Lines 183-191, this paragraph is nothing to do with results. I would suggest to delete them or move to somewhere else.

Is there a clear definition of “effective parameters” shown in Figure 2?

Line 216, any particular reason for choosing Chitgar station to be presented in Figs 3-8?

Can you explain how the RMSE in Table 1 were calculated? What is the observed value and the prediction value used in the RMSE formula?

Line 269, there’s no clear definition of “first” and “second” dataset

Figures 19 and 20, what does “Main map” mean?

The title of first column in Table 7 was “delay”, suggest changed to “delay (days)”

Line 302, does the 94.3% refer to the R2 0.94 in Table 7?

The term “data mining” didn’t appear anywhere else in the manuscript until lines 322-326, is it necessary to be mentioned in the discussions and conclusions?

Is section 5 “future directions” necessary to be an independent section, or the authors may consider to merge it into “discussions and conclusion”

Author Response

Response to Reviewer 2 Comments

Point 1: The Introduction was too heavy (lines 45 -152 and account for almost 1/3 of content in the manuscript), and the worst part is after the long introduction, the value and contributions of this study couldn’t be identified clearly

Response 1: The major contributions of this research presented in sections 5 and line 457 to 468:

The major contributions of this research are as follows:

·           A comparative study of machine learning methods including NARX, ANN, GWR and SVR has been employed for air pollution prediction and the NARX finally selected as the optimum one.

·           The research has improved the efficiency of the machine learning method employed based on filtering the existing noise in both the meteorological and air pollution data as well as predicting the missed meteorological data.

·           This research has proposed a novel approach for air pollution prediction in urban areas based on both stationary and non-stationary pollution sources using machine learning and statistical methods.

·           Fourier series has been implemented for monthly missed meteorological data and spline methods employed for daily and weekly missed meteorological data.

·           The effective parameters for air pollution prediction have been determined in this research.

Point 2:  Please consider revising section 1 thoroughly

Response 2: To summarize the introduction, we have removed some of the following unnecessary sections:

Lines 50 to 55 (in last version):

“Air pollution has several sources including urbanization, development of cities, increased population, and development of industrial activities, increased use of fossil fuels, insufficiency of effective transportation systems, low quality of fuel, and traffic congestion all have caused daily emission of an extensive amount of pollutants to mention some of the main factors to consider. Air pollution is a growing problem worldwide, hence, air-pollution prediction is becoming one of the managerial solutions to prevent and/or reduce its destructive implications.”

Lines 67 to 73 (in last version):

“The sources of PM2.5 emission include different types of combustion (such as motor vehicles, plants, and burning the woods) and specific industrial processes. These particles are either directly emitted or formed in the atmosphere in the form of secondary pollutants. The sources of PM2.5 emission are generated from burning the woods, imperfect combustion in hydrocarbon compositions, and combustion of diesel fuels, industrial and agricultural units, un-asphalted roads, construction activities, and nongaseous pollutants. Therefore, it seems necessary to predict PM10 and PM2.5 pollutants using the appropriate methods.”

Lines 86 to 89 (in last version):

“The sources of PM2.5 emission include different types of combustion (such as motor vehicles, plants, and burning the woods) and specific industrial processes. These particles are either directly emitted or formed in the atmosphere in the form of secondary pollutants. The sources of PM2.5 emission are generated from burning the woods, imperfect combustion in hydrocarbon compositions, and combustion of diesel fuels, industrial and agricultural units, un-asphalted roads, construction activities, and nongaseous pollutants. Therefore, it seems necessary to predict PM10 and PM2.5 pollutants using the appropriate methods.”

Lines 145 to 152 (in last version):

“Due to problems with the deterministic methods, some researchers have used statistical models for modeling the air-pollution. Developing the stochastic models for qualitative modeling of air-pollution involves a number of complexities. This factor engenders higher dependence of this model on precise analysis of emission as well as meteorology that in turn have significant stochasticity. The statistical methods using the existing data on meteorology and air-pollution and their statistical relationship, are considered as a more efficient strategy for prediction of the air-pollution concentration, as their profitability has been proved by studies for short-term prediction of air-pollution using statistical method [20].”

Point 3:  and highlight the objectives of this study with dot points at the end of section.

Response 3: Line 165 to 169 (in new version) is corrected:

The aims of this research are as follows:

Selecting the best      statistical model and its improvement for air-pollution prediction,

Selecting the best      refinement method for      air-pollution and meteorological data in order to predict the missed data      and filter the noise of the data, and

Determination of the most      influencing parameters in air pollution prediction.”

Point 4:  More structured and detailed and descriptions of the original 10-year measurements dataset (both air quality and Met data) used in this study are expected in section 2 “Material and Methods”.

Response 4: Lines 192 to 213 (in last version) in Section 3 are transmitted to lines 180 to 222 (in new version) in Section 2.

Point 5:  There’s no mention to data QA/QC processes, did they have any QA/QC done before releasing to the users?

Response 5: Lines 220 and 221 (in new version) have been added:

“The meteorological and air pollution data have been quality controlled by the concerned organizations.”

Point 6:  Are the minutes or hourly or daily data used in this study?

Response 6: its daily, in line 214 ( in new version) is said.

Point 7:  Is it possible to add a map showing the stations where the data are used?

Response 7: figure5 (in new version) has been added

Point 8:  There’s a well-designed flowchart shown in Figure 1, may I suggest the authors to follow this flowchart and make some dot points to organise all the information related to data acquisition, data refinement, interpolation, noise elimination etc. to help reader to follow.

Response 8: in lines 230 to 250 ( in new version) are said.

Point 9:  Also, there are some data descriptions in the current section 3 (lines 205-213), I would suggest moving all the information related to data into section 2.

Response 9: Lines 205 to 213 (in last version) in Section 3 are transmitted to lines 212 to 222 (in new version) in Section 2.

Point 10:  These are lots of interesting findings and discussions in the 3rd section “Results”, so may I suggest to authors to add at least two sub-sections to accommodate results for 1) data refinement, 2) prediction of air pollution.

Response 10: The results section 3 is divided into two sections:

3.1. Data preparation and refinement

3.2. Air pollution prediction

Point 11:  Lines 162-163, “these data are not error-free”, does this mean the data were not QA/QC before they can be released?

Response 11: yes

Point 12:  Line 172 “These filters are better than ordinary Finite Impulse Response”, any reference to support this?

Response 12: Reference has been added in line 240 and 242 (in new version).

Point 13:  Lines 183-191, this paragraph is nothing to do with results. I would suggest to delete them or move to somewhere else.

Response 13: They are case study.

Point 14:  Is there a clear definition of “effective parameters” shown in Figure 2?

Response 14: The following definition is added in lines 200 to 202(in new version)

“Effective parameters are parameters that changes in location and time which cause changes in the concentration of air pollution.”

Point 15:  Line 216, any particular reason for choosing Chitgar station to be presented in Figs 3-8?

Response 15: Lines 277 to 278 (in new version) have been added.

“Chitger station illustrated in Figure 5 are presented as examples of the total data collected.”

Point 16:  Can you explain how the RMSE in Table 1 were calculated? What is the observed value and the prediction value used in the RMSE formula?

Response 16: Lines 250 to 260 (in new version) have been added:

After each method is implemented and modeled, the results should be validated and compared. Therefore, in this research, two parameters of the coefficient of determination and root mean square error using cross-validation method are used to evaluate the results. The determination coefficient shows the correlation between the observed values and the calculated values which is always between 0 and 1, the value of one representing a complete correlation between the observed values and the calculated values, and the zero value represents the independence of the observed values and the values calculated. The coefficient of determination and the root mean square error have been calculated using Eqs. 1 and 2 is [27], [28].

                                                                                  ,

(1)

,

(2)

where N is the number of observations,  is the observed parameter, Pi is the calculated parameter,   is the mean of the observation parameter,  is the average of the calculation parameter,  is the standard deviation of the observations and  is the standard deviation of the calculation.

”.

And lines 322 and 323 ( in new version) have been added:

“where RMSE is calculated according to Eq. (2).”

Point 17:  Line 269, there’s no clear definition of “first” and “second” dataset

Response 17: In lines 345 and 346 ( in new version) mentioned:

“first dataset (pollutants have not been refined) and second dataset (pollutants have been refined by Savitzky-Golay filter)”.

Point 18:  Figures 19 and 20, what does “Main map” mean?

Response 18: its observation map and it is corrected in figure 22 and 23( in new version).

Point 19:  The title of first column in Table 7 was “delay”, suggest changed to “delay (days)”

Response 19: done

Point 20:  Line 302, does the 94.3% refer to the R2 0.94 in Table 7?

Response 20: In lines 399 (in new version), the following sentence is corrected

“This research will have a significant RMSE of 9.98(µg/m3)”

Point 21:  The term “data mining” didn’t appear anywhere else in the manuscript until lines 322-326, is it necessary to be mentioned in the discussions and conclusions?

Response 21: In lines 424 to 436 (in new version) we changed the “Data Mining” to “statistical method”.

Point 22:  Is section 5 “future directions” necessary to be an independent section, or the authors may consider to merge it into “discussions and conclusion”

Response 22: done

Reviewer 3 Report

Section 4. Discussion and Conclusions should be sparated in two sections:

4. Discussion

5. Conclusions

Author Response

Response to Reviewer 3 Comments

Point 1: Section 4. Discussion and Conclusions should be separated in two sections:

4. Discussion

5. Conclusions

Response 1: We separated these two segments from line 423 to 491
